# Peroxisomal compartmentalization of amino acid biosynthesis reactions imposes an upper limit on compartment size

Ying Gu [1,2] ✉, Sara Alam [1,2,3] & Snezhana Oliferenko [1,2] ✉

Cellular metabolism relies on just a few redox cofactors. Selective compartmentalization may prevent competition between metabolic reactions requiring the same cofactor. Is such compartmentalization necessary for optimal cell function? Is there an optimal compartment size? Here we probe these fundamental questions using peroxisomal compartmentalization of the last steps of lysine and histidine biosynthesis in the fission yeast *Schizosaccharomyces japonicus*. We show that compartmentalization of these $NAD^+$ dependent reactions together with a dedicated $NADH/NAD^+$ recycling enzyme supports optimal growth when an increased demand for anabolic reactions taxes cellular redox balance. In turn, compartmentalization constrains the size of individual organelles, with larger peroxisomes accumulating all the required enzymes but unable to support both biosynthetic reactions at the same time. Our reengineering and physiological experiments indicate that compartmentalized biosynthetic reactions are sensitive to the size of the compartment, likely due to scaling-dependent changes within the system, such as enzyme packing density.

In eukaryotes, many metabolic pathways are partially or fully compartmentalized in subcellular organelles. Compartmentalization of enzymes and their reactants optimizes reaction flux within multiple metabolic networks and manages toxic effects of reaction by-products[1]. The single membrane-bound peroxisomes exhibit great functional versatility. Despite peroxisomes sharing common biogenesis mechanisms and protein import machinery[2], their enzymatic repertoire is highly variable among species, individual cell types and under different physiological conditions. For instance, in addition to well-established oxidative metabolism functions, such as fatty acid (FA) β-oxidation and hydrogen peroxide removal, the budding yeast peroxisomes participate in the biosynthesis of the amino acid L-lysine[3]. Peroxisome-like organelles, glycosomes, in the Trypanosoma parasites, house enzymes for glycolysis, glycerol metabolism, pyrimidine and purine metabolism and ether-linked lipid synthesis[4–7]. During organismal development, peroxisomes may alter their functions by compartmentalizing distinct sets of enzymes. For instance, specialized peroxisomes, glyoxysomes, participate in converting storage lipids to carbohydrates in Arabidopsis seedlings, whereas in mature plants, peroxisomes harbouring a different repertoire of enzymes collaborate with other organelles in supporting photorespiration[8,9].

Peroxisomes are a distributed organelle network, typically present as many individual compartments. Peroxisomes can arise de novo through the endoplasmic reticulum (ER) derived pre-peroxisomal vesicles[10]. Alternatively, they originate through fission of pre-existing mature peroxisomes, which is mediated by Pex11 that remodels the peroxisomal membrane and promotes activation of the dynamin-dependent membrane scission[11–14]. As peroxisomes grow and mature, cargo import into the peroxisomal matrix takes place. Delivery of proteins across the peroxisomal membrane is mediated by two cargo receptors binding to the specialized cis-motifs on cargo proteins – the peroxisomal targeting signal 1 (PTS1)-type receptor Pex5 and the

[1]The Francis Crick Institute, 1 Midland Road, London NW1 1AT, UK. [2]Randall Centre for Cell and Molecular Biophysics, School of Basic and Medical Biosciences, King's College London, London SE1 1UL, UK. [3]Present address: Medical Research Council London Institute of Medical Sciences, Du Cane Road, London W12 0NN, UK. ✉e-mail: ying.1.gu@kcl.ac.uk; snezhka.oliferenko@crick.ac.uk

PTS2-type receptor Pex7[15,16]. Changes in peroxisome size and number have been associated with environmental stimuli in many organisms[17–22] and human disease[23–25].

The fission yeast *Schizosaccharomyces japonicus* provides an attractive stripped-down model for metabolic compartmentalization in peroxisomes. The fission yeast clade has lost the peroxisomal FA β-oxidation system. In fact, all fission yeasts lack a functional glyoxylate cycle, required for the conversion of acetyl-coenzyme A (CoA) generated by FA β-oxidation into four-carbon biosynthetic precursors, and so cannot use FAs as a carbon source[26,27]. Of note, the Lys3 and His2 enzymes catalysing the last steps of lysine and histidine biosynthesis, respectively, carry the C-terminal PTS1-like sequences[26,27], suggesting that these metabolic reactions may take place in peroxisomes. *S. japonicus* grows at the cell tips and divides in the middle, maintaining a polarized pill-shape morphology throughout its vegetative growth[28]. *S. japonicus* maintains a large cell size and divides fast in the nutrient-rich medium but scales down its cell volume when switched to the minimal medium, where cells are forced to synthesize all amino acids[28].

Here, we show that peroxisomal compartmentalization of amino acid biosynthesis facilitates metabolic adaptation of *S. japonicus* to nutrient-poor conditions. We further demonstrate that such a compartmentalization imposes an upper limit on peroxisome size, beyond which the efficiency of amino acid biosynthesis reactions competing for the same nicotinamide adenine dinucleotide NAD$^+$ cofactor is critically decreased.

## Results

### The glycerol-3-phosphate dehydrogenase Gpd2 provides NAD$^+$ for histidine and lysine biosynthesis in peroxisomes

*S. japonicus* does not respire oxygen, unlike *S. pombe*, but grows fast and maintains high energy content. Rather than using the electron transport chain, it relies on the reduction of dihydroxyacetone phosphate (DHAP) to glycerol-3-phosphate (G3P) by the cytosolic glycerol-3-phosphate dehydrogenase Gpd1 to oxidize NADH, to sustain purely fermentative growth[29]. *S. japonicus* but not *S. pombe* cells lacking Gpd1 exhibit severe growth defects and cell shape abnormalities in the rich yeast extract with supplements (YES) medium and are unable to grow at all in the Edinburgh minimal medium (EMM) (Fig. 1a, c, Supplementary Fig. 1a and ref. 29). The fission yeast genomes encode an additional NADH-dependent glycerol-3-phosphate dehydrogenase, Gpd2[26,30]. *S. japonicus* gpd2Δ cells exhibited normal morphology, but their rate of growth was slightly attenuated in the rich medium (Fig. 1b–d). Interestingly, the mutant cells rounded up and fully arrested growth upon transfer to EMM (Fig. 1b–d). The loss of cellular polarity leading to virtually spherical cells is typical for growth arrest in *S. japonicus* that relies on its geometry for cell division[28] and can be used as a proxy for estimating population growth with a single-cell resolution. The inability of Gpd2-deficient cells to grow in the minimal medium was conserved in *S. pombe*, consistent with a previous report (Supplementary Fig. 1a, b and ref. 30).

Gpd2 was enriched in peroxisomes in *S. japonicus* (Fig. 1e, left panel, and Supplementary Fig. 1c). As expected from the presence of the C-terminal PTS1-type peroxisome targeting signals, Lys3 and His2 enzymes catalysing the last reactions of lysine and histidine biosynthesis respectively, also localised to peroxisomes (Fig. 1e, middle and right panels and Supplementary Fig. 1c). Both reactions require NAD$^+$, suggesting a possible role for Gpd2 in re-oxidizing NADH to sustain lysine and histidine synthesis in peroxisomes (Fig. 1f). Indeed, growth inhibition of *S. japonicus* gpd2Δ cells was rescued only when the minimal medium was supplemented with both lysine and histidine (Fig. 1d).

Growth in minimal medium requires cells to synthesize all amino acids. Since many anabolic reactions require NAD$^+$, it may challenge the cellular redox balance. Indeed, the NAD$^+$/NADH ratio dramatically decreased when *S. japonicus* cells grown in the rich YES medium were transferred to the minimal EMM (Fig. 1g). In non-respiring *S. japonicus*, which has very little ubiquinone[29,31], excessive NADH may lead to an increased production of reactive oxygen species via the NADH:ubiquinone oxidoreductase[32,33]. Consistent with a shift in cellular redox status, the switch to the minimal medium elicited transient upregulation of cytosolic catalase Ctt1, a hallmark of oxidative stress response[34] (Supplementary Fig. 1d, e). Such a dramatic upregulation did not occur in *S. pombe* (Supplementary Fig. 1e, left panel). This result indicates that redox balancing mechanisms in *S. japonicus* may struggle to meet the demands of increased anabolism.

The catalase upregulation was even more pronounced in *S. japonicus* cells lacking Gpd2 (Fig. 1h and Supplementary Fig. 1f, upper panel). Supplementing the growth medium with lysine and histidine restored the kinetics of catalase regulation in gpd2Δ cells to wild type levels (Fig. 1h and Supplementary Fig. 1f, bottom panel). Gpd1 also exhibited major upregulation in *S. japonicus* upon the shift to the minimal medium, in line with its function in redox balancing (Supplementary Fig. 1e, right panel). In fact, an ectopic copy of Gpd1 fused to PTS1, Gpd1$^{PTS1}$, and expressed from the gpd2 promoter, was able to rescue polarity scaling and growth in *S. japonicus* gpd2Δ cells upon shift to the minimal medium (Fig. 1i, j). Taken together, these results suggest that beyond the role in synthesizing glycerol-3-phosphate, the glycerol-3-phosphate dehydrogenases Gpd1 and Gpd2 execute redox balancing in the cytosol and inside peroxisomes, respectively.

### Selective compartmentalization of Lys3 and His2 facilitates cellular redox balance to meet the demand for de novo amino acid biosynthesis

To understand why cells may benefit from selective compartmentalization of certain amino acid biosynthesis reactions, we sought to create scenarios, where NAD$^+$ dependent Lys3 and His2 enzymes normally present in peroxisomes were delocalized to the cytosol in the presence or absence of the glycerol-3-phosphate dehydrogenases.

*S. japonicus* cells lacking Pex5 but not Pex7 failed to import Lys3 and His2 into peroxisomes. Similarly, the ectopically expressed Gpd1$^{PTS1}$ delocalized to the cytosol in pex5Δ cells (Supplementary Fig. 2a). The *S. japonicus* Gpd2 harbours an N-terminal PTS2; as expected, it largely redistributed to the cytoplasm in the absence of Pex7 (Supplementary Fig. 2a). Either the deletion or point mutations of PTS2 (see Methods) fully delocalized Gpd2 to the cytoplasm and arrested growth of mutant cells upon transfer to the minimal medium (Supplementary Fig. 2b), underscoring functional importance of peroxisomal targeting of this protein. Interestingly, Gpd2 import to peroxisomes was also partially disrupted by Pex5 loss of function (Supplementary Fig. 2a), consistent with previous reports in mammalian cells, plants, *Trypanosoma* and the fungal plant pathogen *Ustilago maydis*, where Pex5 was shown to serve as a Pex7 co-receptor for the PTS2 cargo translocation across the peroxisomal membrane[35–39].

We wondered if releasing Lys3 and His2 from peroxisomes into cytosol by disabling Pex5 would allow lysine and histidine biosynthesis utilizing cytosolic NAD$^+$. Indeed, not only pex5Δ but also pex5Δ gpd2Δ double mutant *S. japonicus* cells were capable of growth in the minimal medium, albeit at a reduced growth rate (Fig. 2a, b and Supplementary Fig. 2c). We also observed a similar rescue of growth without amino acid supplementation when we deleted pex3, which fully abolishes peroxisome biogenesis, or pex3 in combination with gpd2 (Supplementary Fig. 2c). Interestingly, pex3 deletion also noticeably improved the growth of gpd1Δ cells in the rich YES medium but not in the minimal medium where cells are forced to synthesize all amino acids (Fig. 2c; see Fig. 1c for comparison). This result argues that although in principle, Gpd2 can substitute for Gpd1 in supporting redox balance in the cytosol, the correct dosage of NAD$^+$ regenerating enzymes is critical when cellular anabolic demands are increased.

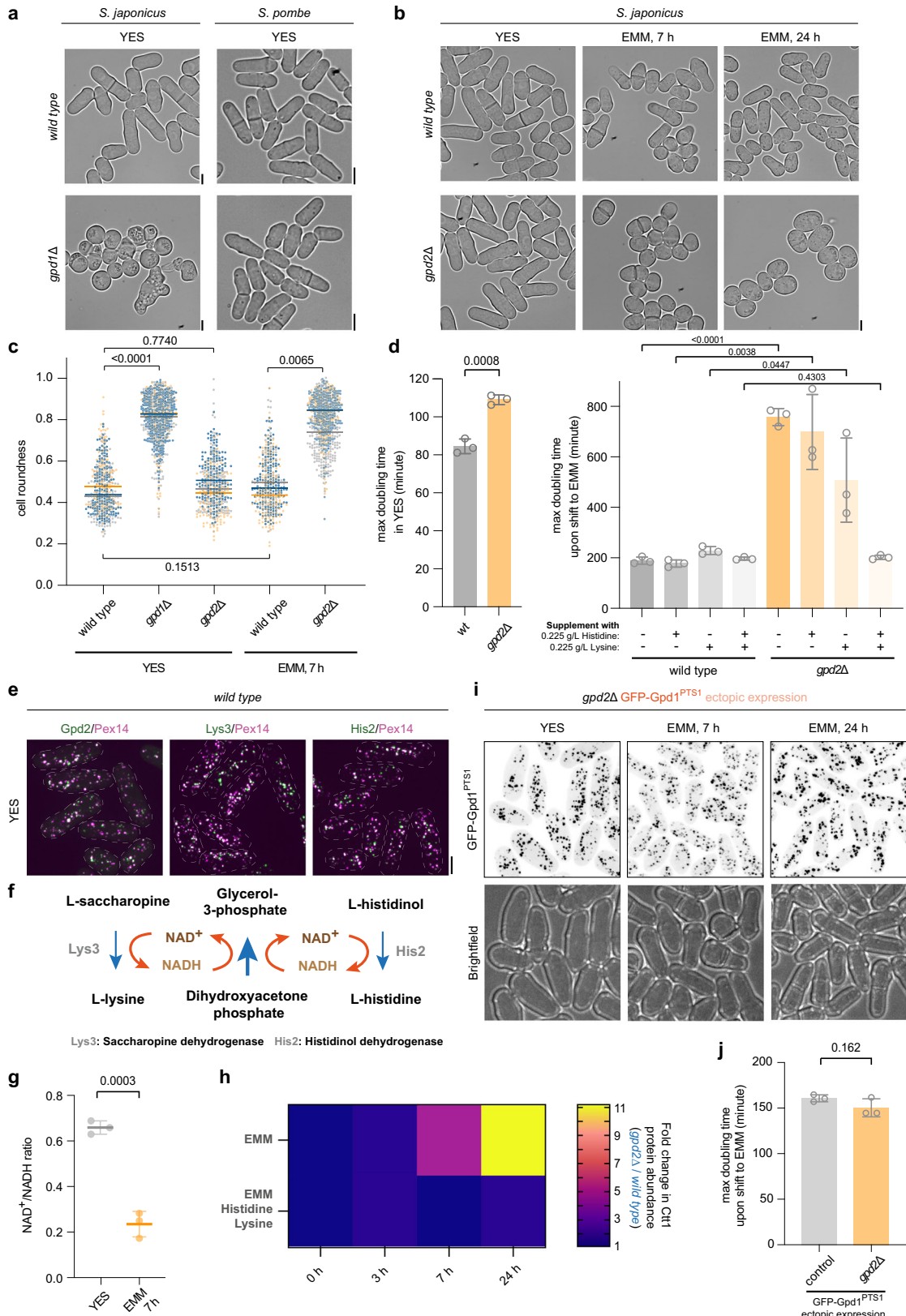

Gas chromatography-mass spectrometry (GC-MS) measurements showed significant disbalance in the intracellular levels of many amino acids in *pex5Δ S. japonicus* cells grown in the minimal medium, as compared to the wild type (Fig. 2d). The levels of many amino acids synthesized in the cytosol were decreased (Fig. 2d, e). Interestingly, the amino acids the synthesis of which requires the cytosolic NAD⁺ or

NADPH-dependent steps were particularly affected. For instance, although leucine and valine share the same precursor, pyruvate[40], only leucine biosynthesis utilises a cytosolic NAD⁺-dependent enzyme, Leu1[26,27,40–42]. We observed that leucine but not valine levels were decreased in *pex5Δ* cells, as compared to the control (Fig. 2d, e). Similarly, the loss of peroxisomal compartmentalization led to the

**Fig. 1 | Peroxisomal glycerol-3-phosphate dehydrogenase Gpd2 enables de novo lysine and histidine biosynthesis. a** Micrographs of wild type and *gpd1Δ S. japonicus* and *S. pombe* cells grown in the yeast extract with supplements (YES) medium. **b** Micrographs of wild type and *gpd2Δ S. japonicus* cells in YES and following a switch to the Edinburgh minimal medium (EMM) medium for 7 and 24 h. **c** Cell morphology profiles of wild type (*n* = 180, 187 and 172 cells grown in YES; *n* = 152, 144 and 185 cells in EMM), *gpd1Δ* (*n* = 283, 200 and 377 cells in YES) and *gpd2Δ* (*n* = 154, 160 and 220 cells in YES; *n* = 230, 326 and 272 cells in EMM) *S. japonicus* cultures in indicated media. Bars represent medians. **d** Growth rates of wild type and *gpd2Δ S. japonicus* cultures grown in YES (left) and post medium switch to EMM, with or without supplementation with indicated amino acids (right). **e** Colour overlays of maximum Z-projection spinning disk confocal images of *S. japonicus* cells co-expressing Pex14-mCherry (magenta) and Gpd2-mNeon-Green, GFP-Lys3 or GFP-His2 (green). **f** Illustration of the last steps in L-lysine and L-histidine biosynthesis pathways, catalysed by Lys3 and His2. Both reactions reduce the cofactor nicotinamide adenine dinucleotide NAD⁺ to NADH, which can be re-oxidised to NAD⁺ via glycerol-3-phosphate synthesis from dihydroxyacetone. **g** Total cellular NAD⁺/NADH ratios of *S. japonicus* grown in YES and 7 h post-switch to EMM. **h** Catalase Ctt1-mNeonGreen protein abundance in *gpd2Δ S. japonicus* cells normalised to the wild type, at 0, 3, 7 and 24 h time points, post-switch from YES to EMM or EMM supplemented with L-lysine and L-histidine. Heatmap shows ratios between populational means of average cell fluorescence intensities. **i** Maximum Z-projection spinning disk confocal images of *gpd2Δ S. japonicus* cells expressing an extra copy of GFP-tagged Gpd1 with the PTS1, in indicated media. **j** Growth rates of *S. japonicus* wild type and *gpd2Δ* cells expressing GFP-tagged Gpd1$^{PTS1}$, post-switch to EMM. **a, b, e, i** Scale bars represent 5 μm. **c, d, g, j** *p*- values are derived from two-tailed unpaired *t*-test. **c, d, g, h, j** Values are derived from three biological replicates. **d, g, j** Bars represent mean values ± SD. Source data are provided as a Source Data file.

reduction in histidine levels (Fig. 2d). Supplementation of histidine improved but did not fully rescue the proliferation defect of *pex5Δ* cells (Supplementary Fig. 2d).

Strikingly, the intracellular levels of lysine were profoundly increased in *pex5Δ* cells (Fig. 2d, e). In budding yeast, the lack of the peroxisomal E3 ubiquitin ligase Pex12, which delocalizes the PTS1 cargo to the cytosol, leads to transcriptional upregulation of several genes in the lysine biosynthesis pathway (Supplementary Fig. 2e)[43]. The upregulation was proposed to be a compensation response regulated by the transcription activator Lys14. However, Lys14 is not conserved outside of Saccharomycetaceae, and we did not observe dramatic changes in Lys2, Lys4, Lys9 and Lys12 protein levels in *pex5Δ S. japonicus* mutants (Supplementary Fig. 2f, g). Of note, the *S. japonicus* Lys4 was confined to mitochondria, unlike its orthologs in *S. pombe* (Supplementary Fig. 2h) or *S. cerevisiae*[44–47]. Our results suggest that upregulation of lysine production by delocalized Lys3 could drive higher flux through the entire lysine biosynthesis pathway, which includes several NAD⁺ and NADPH-dependent reactions, potentially competing with other metabolic reactions and causing amino acid disbalance (Fig. 2d, e).

Since growth in the minimal medium taxes the intracellular supply of NAD⁺ (Fig. 1g), we wondered if providing cells with more NADH-oxidizing activity could rescue poor growth of *pex5Δ* cells in EMM. We made use of an ectopically expressed PTS1-carrying version of Gpd1 (Gpd1$^{PTS1}$) that localized to the cytosol upon loss of Pex5 cargo receptor (Supplementary Fig. 2a). Indeed, *pex5Δ* cells expressing ectopic Gpd1$^{PTS1}$ proliferated normally in the minimal medium (Fig. 2f). Although the amino acid profile of *pex5Δ* cells expressing ectopic Gpd1$^{PTS1}$ was now more similar to the control, we did not observe a full rescue (Fig. 2g). In particular, lysine levels remained high, suggesting that the compartmentalization of the last step of lysine biosynthesis in peroxisomes was important for controlling the activity of this biosynthetic pathway.

### Abnormal peroxisome compartment architecture impedes amino acid biosynthesis

We noticed that the *S. japonicus* cells lacking the peroxisomal fission factor Pex11[48] struggled to adapt to growth in the minimal EMM medium (Fig. 3a, b, compare with Fig. 1b, c). Supplementation of the medium with either lysine or histidine significantly restored growth and normal cell morphology in *pex11Δ* cultures (Fig. 3b). Hyper-accumulation of the cytosolic catalase Ctt1, indicative of prolonged oxidative stress response was also alleviated in *pex11Δ* cells when the growth medium was supplemented with either lysine or histidine (Fig. 3c, d). These results suggested that although Lys3 or His2-mediated reactions could work in principle, they were not sufficiently productive when the demand for lysine and histidine biosynthesis was high. Interestingly, some *pex11Δ* cells were able to return to polarized growth after prolonged incubation in the EMM (see the 24 h time-point

in Fig. 3a). The ability to grow in the minimal medium coincided with a virtual disappearance of peroxisomes in these cells (Supplementary Fig. 3a, b), likely due to a failure in the inheritance of abnormally large peroxisomes by some daughter cells.

Targeting of ectopic Gpd1$^{PTS1}$ to peroxisomes was unable to rescue the growth arrest of *pex11Δ* cells (Fig. 3e), suggesting that the insufficient dosage of the redox balancing Gpd2 enzyme was not responsible for functional lysine and histidine auxotrophy in this genetic background.

We concluded that other parameters pertaining to peroxisome compartment architecture, such as the size and/or the number of individual peroxisomes may dictate the productivity of lysine and histidine biosynthesis.

### Optimal lysine and histidine biosynthesis imposes an upper limit on peroxisome size

In line with the function of Pex11 in promoting peroxisome fission[48], we observed fewer peroxisomes that were larger in *pex11Δ S. japonicus* cells, as compared to the wild type. The peroxisome density (a parameter describing the number of compartments per unit area in maximum projection images) was $0.59 \pm 0.12/\mu m^2$ (mean±SD) in YES, increasing to $0.74 \pm 0.15/\mu m^2$ in EMM-grown wild type cells. In the absence of Pex11, peroxisome density was lower in both conditions ($0.34 \pm 0.12/\mu m^2$ in YES and $0.33 \pm 0.10/\mu m^2$ in EMM). The estimated median peroxisome volume in the wild type (~$4.56 \times 10^{-3}$ μm³) did not change significantly between rich and minimal media. However, this parameter was increased to $6.36 \times 10^{-3}$ μm³ in YES and further to $9.64 \times 10^{-3}$ μm³ in EMM in *pex11Δ* cells. Thus, peroxisome size in Pex11-deficient cells grown in the minimal medium was ~2.1x larger than in the wild type (Fig. 4a–c).

To investigate the role of peroxisome architecture further, we identified an *S. japonicus* mutant harbouring a single amino acid substitution from tryptophan to alanine at amino acid residue 224 in Pex5 (Pex5-W224A, see sequence alignment of fission yeast Pex5 proteins in Supplementary Fig. 4a). In the rich YES medium, the *pex5-W224A* cells exhibited higher number of peroxisomes ($1.03 \pm 0.15/\mu m^2$), coinciding with reduction in individual compartment size ($2.37 \times 10^{-3}$ μm³) (Fig. 4a–c). Upon transfer to the minimal medium, these parameters ($0.75 \pm 0.13/\mu m^2$ and $5.01 \times 10^{-3}$μm³) were more comparable to the wild type. Despite the differences in compartment organisation, the peroxisomal import of Lys3, His2 or Gpd2 was not affected (Supplementary Fig. 4b). *pex5-W224A S. japonicus* mutant cells could grow in the minimal medium, indicating that they were capable of lysine and histidine biosynthesis (growth rate of *pex5-W224A* cells in EMM was $149.2 \pm 12.3$ min versus $162.4 \pm 15.4$ min for wild type cells (mean±SD, *n* = 3, *p*-value = 0.3073, two-tailed unpaired *t*-test)).

Of note, the introduction of *pex5-W224A* mutation to *pex11Δ* genetic background decreased the size ($5.09 \times 10^{-3}$ μm³ in EMM) of individual peroxisomes, which was now comparable to that of the

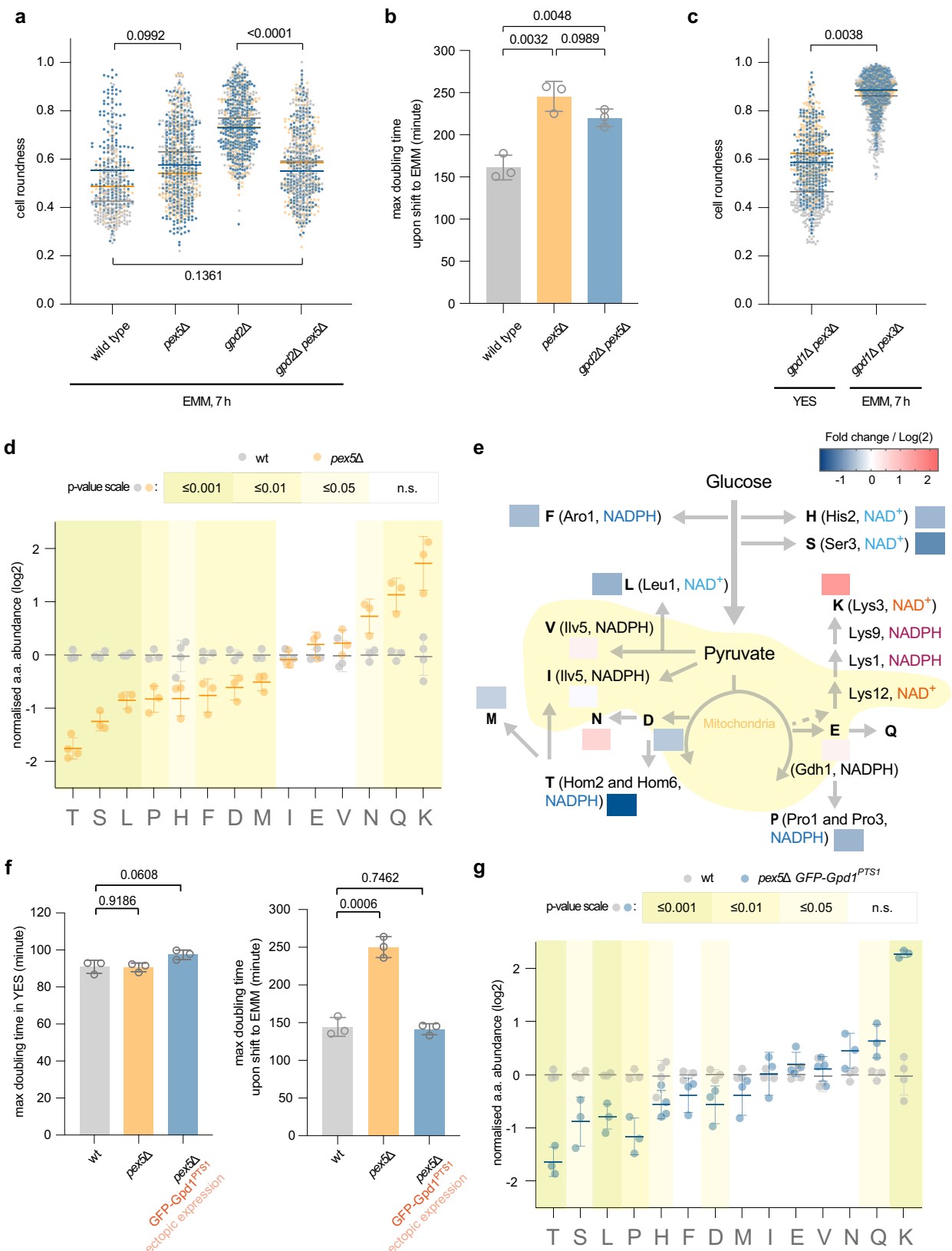

**d** wt ● *pex5Δ* ●

p-value scale ● ● : ≤0.001 ≤0.01 ≤0.05 n.s.

wild type (Fig. 4a–c). Strikingly, *pex11Δ pex5-W224A* double mutant cells exhibited the wild-type like growth pattern in the minimal medium (Fig. 4d). Pex11 has been proposed to function as a membrane channel in vitro[49]. The fact that the *pex5-W224A* mutation rescues the inability of Pex11-deficient cells to grow in the minimal medium indicates that the compartment architecture rather than

Pex11 function(s) per se defines the amino acid biosynthetic capacity of peroxisomes.

Interestingly, using 3D segmentation analysis in the wild type and the peroxin mutants described above, we observed that on a population level the peroxisome size exhibited positive correlation with an average intensity of peroxisomal proteins (see Fig. 4e for

**Fig. 2 | Compartmentalization of Lys3 and His2 enzymes in peroxisomes satisfies the redox demand for balanced amino acid biosynthesis. a** Cell morphology profiles of *S. japonicus* cells of wild type (*n* = 114, 179 and 205 cells), *pex5Δ* (*n* = 234, 270 and 299 cells), *gpd2Δ* (*n* = 207, 291 and 241 cells) and *gpd2Δ pex5Δ* (*n* = 239, 236 and 232 cells) in the Edinburgh minimal medium (EMM) for 7 h post-shift from the yeast extract with supplements (YES) medium. **b** Growth rates of *S. japonicus* cultures of indicated genotypes post medium switch to EMM. **c** Cell morphology profiles of *gpd1Δ pex3Δ S. japonicus* cells under indicated conditions (*n* = 302, 258 and 253 cells in respective biological replicate grown in YES; *n* = 383, 374 and 471 cells in EMM). **d** Fold changes of amino acids abundances (expressed as log2) in *pex5Δ S. japonicus* cells normalised to the wild type, sampled at 7 h post-shift to EMM. Amino acids are denoted by one-letter symbols. **e** Illustration of amino acid biosynthesis pathways derived from glycolysis and the tricarboxylic acid cycle (TCA) cycle. Pathway logics are indicated by grey lines with arrows. Please note that the TCA pathway operates in a bifurcated configuration in *S.*

*japonicus*[29]. Fold changes of individual amino acid abundance in *pex5Δ S. japonicus* cells normalised to the wild type shown in **d** are represented as heatmaps. Known enzymes involved in redox reactions utilising either nicotinamide adenine dinucleotide NAD⁺ or NADPH co-factors are indicated next to the respective amino acid products. **f** Growth rates of *S. japonicus* wild type, *pex5Δ* and *pex5Δ* cells expressing GFP-tagged Gpd1[PTS1] grown in YES (left) and post-shift to EMM (right). **g** Fold changes of amino acids abundances (expressed as log2) in *pex5Δ S. japonicus* cells expressing GFP-tagged Gpd1[PTS1] normalised to the wild type, sampled at 7 h post-shift to EMM. **a–c, f** Values are derived from three biological replicates. *p*-values are derived from two-tailed unpaired *t*-test. **d, g** Values are from two technical repeats of two biological replicates. *p*-values are derived from two-tailed unpaired *t*-test analysis and colour-coded (see the scale above the graphs). **a, c** Bars represent mean population medians. **b, d, f, g** Bars represent mean values ± SD. Source data are provided as a Source Data file.

Gpd2-mNeonGreen and Supplementary Fig. 4c for GFP-Lys3 and GFP-His2). This suggested that the concentration of enzymes could be significantly higher in enlarged peroxisomes in *S. japonicus pex11Δ* cells, as compared to the wild type. The total levels of Lys3, His2, Gpd2 and Gpd1 were comparable between the wild type and *pex11Δ* cells (Supplementary Fig. 4d, e).

An increase in enzyme abundance does not necessarily guarantee a higher biosynthetic output within the compartment[50], especially if several pathways share a common limiting factor. We reasoned that selectively moving either Lys3 or His2 reactions out of the peroxisomal matrix to cytosol will allow us to test if peroxisome size impacted on the productivity of lysine and histidine biosynthesis. To this end, we used a GFP-binding nanobody approach, where we tethered either enzyme to the microtubule- and polar cortex-associated protein Tea1[51] tagged with mCherry and the GFP-binding protein (GBP)[52] (Supplementary Fig. 5a). As a control, we constructed a truncated version of Tea1, which did not interact with microtubules or cell polarity factors and exhibited diffuse cytosolic localization (Tea1[1–407]-mCherry-GBP; Supplementary Fig. 5b). We reasoned that upon the interaction with GFP-tagged Lys3 or His2, the full length Tea1-mCherry-GBP will be able to retain at least a fraction of these enzymes in the cytosol allowing them to use cytosolic NAD⁺, whereas the truncated Tea1[1–407]-mCherry-GBP will be imported into peroxisomes alongside them. The peroxisomal features, such as their size, number and enzyme intensities, were comparable in cells expressing either the full-length Tea1-mCherry GBP or the truncated Tea1[1–407]-mCherry-GBP, both in the wild type or *pex11Δ* genetic background (Supplementary Fig. 5c-f and Supplementary Fig. 5g–j).

The full length Tea1-mCherry-GBP colocalized with GFP-Lys3 or GFP-His2 in puncta that were often enriched at the cell tips in the minimal medium (Fig. 4f and Supplementary Fig. 5k; note that a cell tip localization was more obvious in the case of Lys3.). Gpd2-mTagBFP2 was found at many but not all the puncta (Supplementary Fig. 5l). These results suggested that Tea1 continued to interact with the cellular polarity machinery, and GFP-Lys3 (or GFP-His2) Tea1-mCherry-GBP complexes largely decorated the surface of peroxisomes. Strikingly, the *pex11Δ* cells where either GFP-Lys3 or GFP-His2 were tethered away from the peroxisomal matrix by the full-length Tea1-mCherry-GBP, were able to sustain normal polarized growth upon the shift to the minimal medium, a rescue of the growth arrest displayed by *pex11Δ* cultures (Fig. 4g). In contrast, the introduction of the truncated Tea1[1–407]-mCherry-GBP in either GFP-Lys3 or GFP-His2 expressing *pex11Δ* cells failed to support their growth in EMM (Supplementary Fig. 5m).

We concluded that peroxisome size was an important determinant in supporting the productivity of lysine and histidine biosynthesis reactions competing for the common co-factor NAD⁺ (Fig. 5).

## Discussion

Vegetatively growing *S. japonicus* maintains a large number of peroxisomes. The overall peroxisome biogenesis machinery in this organism is well conserved[2,26], with Pex3 being a critical upstream biogenesis factor, Pex5 and Pex7 serving as PTS1 and PTS2 import cargo receptors (Fig. 2 and Supplementary Fig. 2), respectively, and Pex11 promoting peroxisomal fission (Fig. 4). Interestingly, *S. japonicus* appears to rely on Pex5 to facilitate Pex7-dependent cargo import (Supplementary Fig. 2a). This is similar to many other organisms, including mammals and plants. Of note, the Schizosaccharomyces genomes do not encode the Pex7 co-receptors, used in budding yeast and Neurospora lineages[2].

Of interest, we have identified a Pex5 point mutation (W224A), that modulates peroxisome size in *S. japonicus* (Fig. 4a–c). This amino acid is located in a third, inverted WxxxY/F motif of *S. japonicus* Pex5, which is structurally similar to an inverted WxxxF motif of the budding yeast ortholog. In *S. cerevisiae*, this motif was proposed to facilitate the PTS1-independent import of two FA oxidation enzymes, both of which are absent in fission yeasts[53]. The mechanistic basis underlying the decrease in median peroxisome size in this mutant remains to be elucidated. Although we did not observe deficient import for the abundant enzyme cargoes we have tested (Supplementary Fig. 4b), it is possible that there is a protein(s) important for membrane architecture, transport of which could be affected by the *pex5-W224A* mutation, or this mutation affects another, previously undetected aspect of Pex5 function in peroxisome size control.

Compartmentalization of the NAD⁺ dependent reactions producing lysine and histidine appears to be a critical function for fission yeast peroxisomes. Unlike *S. cerevisiae*, the fission yeasts do not have the malate/oxaloacetate NAD⁺/NADH Mdh3 shuttle[3], and rely fully on Gpd2 to oxidize NADH in this compartment. The release of Lys3 and His2 enzymes into cytosol by genetically ablating Pex5 leads to a pronounced disbalance in cellular concentrations of lysine and histidine, but also a number of other amino acids. Of note, although histidine levels in *pex5Δ S. japonicus* cells drop in comparison to the wild type, the concentration of lysine is increased virtually four-fold. In *S. pombe* and *S. cerevisiae*, an increase in free lysine inhibits the first step in the lysine biosynthesis pathway through the homocitrate synthase (ref. 47,54,55; Supplementary Fig. 2e). Interestingly, the *S. japonicus* homocitrate synthase Lys4 localizes to mitochondria (Supplementary Fig. 2f), unlike its homologues in *S. pombe* and *S. cerevisiae*, which are distributed throughout the cytosol and/or the nucleus (Supplementary Fig. 2h and ref. 44–47). Compartmentalization may shield the *S. japonicus* Lys4 from such an inhibition[56]; it is also possible that it had acquired mutation(s) desensitizing it to inhibition by the end product of the pathway[57].

In the absence of negative feedback over lysine production[47,58,59], peroxisomal compartmentalization of the lysine producer Lys3 may

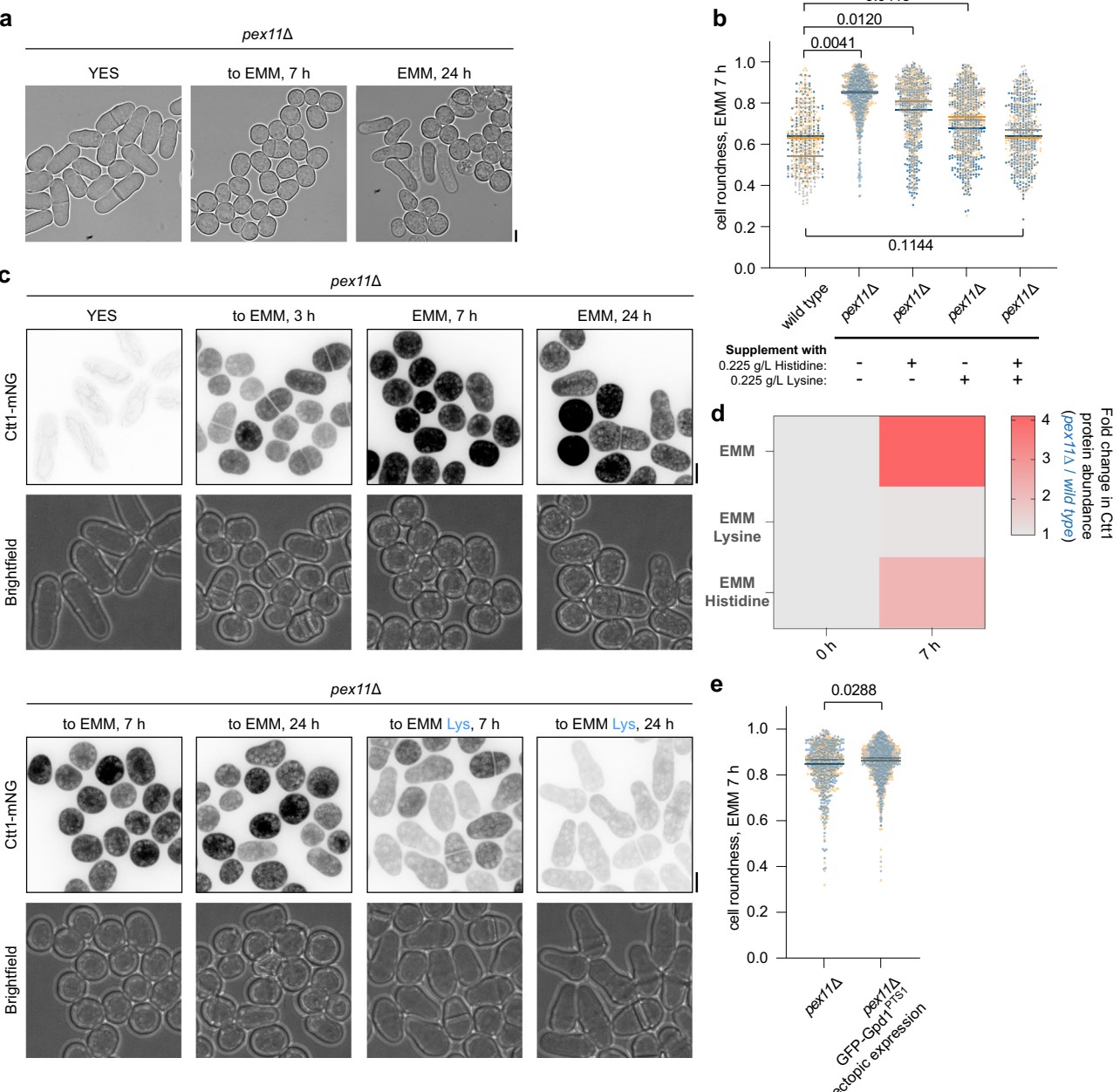

**Fig. 3 | Abnormal peroxisome compartment architecture blocks lysine and histidine biosynthesis. a** Micrographs of *pex11Δ S. japonicus* cells grown in the yeast extract with supplements (YES) and post-shift to the Edinburgh minimal medium (EMM). **b** Cell morphology profiles of wild type and *pex11Δ S. japonicus* cells sampled at 7 h post-shift to EMM or EMM with indicated amino acid supplementation. Wild type grown in EMM, *n* = 249, 163 and 160 cells, respectively; *pex11Δ* in EMM, *n* = 345, 431 and 302 cells, respectively; *pex11Δ* in EMM with histidine, *n* = 278, 272 and 311 cells, respectively; *pex11Δ* in EMM with lysine, *n* = 328, 343 and 322 cells, respectively; and *pex11Δ* in EMM with lysine and histidine, *n* = 168, 261 and 333 cells, respectively. **c** Maximum Z-projection spinning disk confocal images of *pex11Δ S. japonicus* cells expressing Ctt1-mNeonGreen in indicated growth conditions. Supplemented amino acid L-lysine (Lys) is indicated in blue. Brightfield

images are shown underneath the corresponding fluorescence channel images. **d** Catalase Ctt1-mNeonGreen protein abundance in *pex11Δ S. japonicus* cells normalised to the wild type, at time points 0 and 7 h, post-switch from YES to EMM, EMM supplemented with L-lysine or EMM supplemented with L-Histidine. Heatmap shows ratios between populational means of average cell fluorescence intensity. **e** Cell morphology profiles of *pex11Δ* (*n* = 277, 324 and 320 cells, respectively) and *pex11Δ* cells expressing an extra copy of GFP-Gpd1$^{PTS1}$ (*n* = 324, 440 and 376 cells, respectively), sampled at 7 h post-shift to EMM. **a, c** Scale bars represent 5 µm. **b, d, e** Values are derived from three biological replicates. **b, e** Bars represent median values. *p*-values are derived from two-tailed unpaired *t*-test analysis. Source data are provided as a Source Data file.

prevent it from competing for the common NAD$^+$ pool in the cytosol. Of note, although extra GPD activity in the cytosol does partially ameliorate the disbalance in the overall free amino acid profile of *pex5Δ S. japonicus* cells, it boosts the lysine levels even further, emphasizing the importance of peroxisomal sequestration of Lys3 enzymatic activity (Fig. 2d, e, g).

In addition to preventing a highly active enzyme from accessing the common pool of reactants, compartmentalization may offer other benefits. Unusual for dehydrogenases, the histidinol dehydrogenase (His2 in fission yeasts) can only bind NAD$^+$ after interacting with its substrate histidinol[60,61]. It catalyses the synthesis of histidine in two sequential NAD$^+$-dependent oxidations[60,61]. The oxidation from

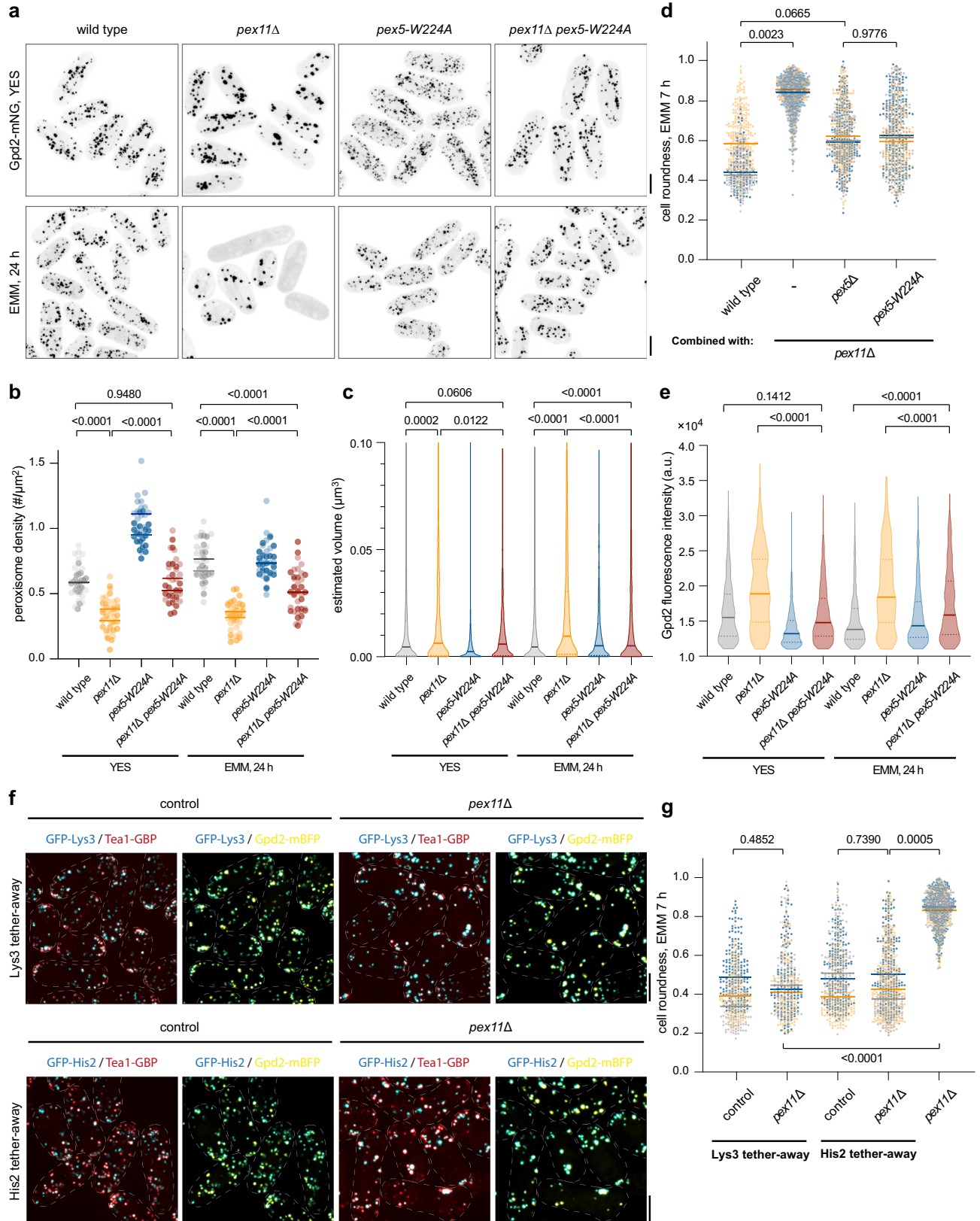

histidinol to histidinaldehyde in the first half-reaction is relatively fast and thermodynamically unfavourable, and the sequential oxidation of histidinaldehyde is slower but irreversible[62]. A dedicated NADH·NAD$^+$ recycling system inside peroxisomes, maintaining a high local concentration of NAD$^+$ may be beneficial to such an enzymatic reaction with suboptimal thermodynamics or kinetics by driving enzymatic flux

towards production. Indeed, we have observed a reduction in histidine levels when His2 is released into the cytosol (Fig. 2).

In addition to a tailored supply of redox factors, cellular compartments may offer different proton environments for metabolic reactions[63,64]. For instance, at least in vitro, production of lysine and histidine by their respective dehydrogenases is favoured in alkaline

**Fig. 4 | Productivity of lysine and histidine biosynthesis relies on optimal peroxisome size. a** *S. japonicus* cells of indicated genotypes expressing Gpd2-mNeonGreen under indicated conditions. **b** Peroxisome density shown in **a.** n = 20 cells per genotype and per condition in biological replicate. **c** Peroxisome size shown in **a.** In YES, wild type, n = 1001 and 952 peroxisomes; *pex11Δ*, n = 463 and 586 peroxisomes; *pex5-W224A*, n = 1633 and 1785 peroxisomes; *pex11Δpex5-W224A*, n = 998 and 1056 peroxisomes. In EMM, wild type, n = 810 and 862 peroxisomes; *pex11Δ*, n = 330 and 320 peroxisomes; *pex5-W224A*, n = 922 and 907 peroxisomes; *pex11Δpex5-W224A*, n = 513 and 624 peroxisomes. **d** Cell morphology profiles of indicated genotypes sampled at 7 h post-shift to EMM. For wild type, n = 284, 105 and 186 cells; *pex11Δ*, n = 346, 307 and 395 cells; *pex11Δ pex5Δ*, n = 274, 236 and 197 cells; *pex11Δpex5-W224A*, n = 229, 225 and 256 cells; **e** Average fluorescence intensities of Gpd2-mNeonGreen within peroxisomes shown in **a.** In YES, wild type, n = 297 and 210 peroxisomes; *pex11Δ*, n = 263 and 300 peroxisomes; *pex5-W224A*, n = 292 and 254 peroxisomes; *pex11Δpex5-W224A*, n = 325 and 234 peroxisomes. In

EMM, wild type, n = 293 and 184 peroxisomes; *pex11Δ*, n = 259 and 239 peroxisomes; *pex5-W224A*, n = 257 and 237 peroxisomes; *pex11Δpex5-W224A*, n = 340 and 213 peroxisomes. **f** *S. japonicus* wild type and *pex11Δ* cells upon the shift to EMM for 24 h. These cells co-express Tea1-mCherry-GBP (red) and Gpd2-mTagBFP2 (yellow) with either GFP-Lys3 (cyan) or GFP-His2 (cyan). **g** Cell morphology profiles of indicated genotypes sampled at 7 h post-shift to EMM. In Lys3 tether-away, control category n = 172, 191 and 129 cells; *pex11Δ* category n = 207, 150 and 147 cells. In His2 tether-away, control category n = 219, 210 and 192 cells; *pex11Δ* category n = 341, 216 and 208 cells. For *pex11Δ*, n = 336, 281 and 306 cells. **a, f** Scale bars represent 5 μm. **b, c, e** Values are derived from two biological replicates. *p*-values are derived from two-tailed unpaired Welch's *t*-test in **b** and Mann–Whitney two-tailed unpaired test in **c, e**. Solid lines represent medians in **b, c, e**. Dotted lines represent the upper and lower quartiles in **c, e. d, g** Values are derived from three biological replicates. *p*-values are derived from two-tailed unpaired *t*-test. Bars represent medians. Source data are provided as a Source Data file.

pH[65,66]. It would be of interest to see if the pH inside fission yeast peroxisomes indeed differs from that in the cytosol, as it was shown in other systems[67], and if potential pH difference could promote or constrain the metabolic flux.

Metabolic compartmentalization has potential trade-offs, such as the energy costs associated with compartment assembly and maintenance, the import and recycling of enzymes, and the transport of solutes. Whereas it is generally agreed that many small molecules can access peroxisomal interior through passive diffusion[68], it remains unclear how NAD(H) is imported into peroxisomes in mammals and yeasts[68]. In principle, NAD(H) could be carried in enzyme-bound form via the peroxisome protein import machinery, with its redox recycling mediated by the peroxisomal redox enzymes[69,70], e.g., Gpd2 in fission yeasts. The energy costs may increase with further metabolic functionality. For instance, peroxisomal FA β-oxidation, lost in fission yeasts but present in many fungi, requires dedicated ATP-dependent transporters for FA import and an antiporter promoting ATP exchange with AMP/ADP across the peroxisomal membrane[71–74].

Subcellular compartmentalization may allow better local management of redox balance risks. For instance, the levels of proline and threonine, which are synthesized in the cytosol in a NADPH-dependent manner decrease in *pex5Δ* cells, as compared to the wild type (Fig. 2d). Yet, there are no differences in the levels of valine and isoleucine, which also require NADPH for their synthesis but are produced in mitochondria. This phenomenon could be related to an increased flux through the entire lysine biosynthesis pathway, since the NADPH-consuming part of it localizes in the cytosol (Supplementary Fig. 2e, f).

Cellular compartmentalization has emerged across all kingdoms of life. Presumably, it provides a universal strategy to modulate flux of biological networks by spatially segregating enzymes and their coupled reactants into organelles, whether enclosed by membranes and proteinaceous shells, or phase separated into condensates[56,75]. Theoretical work has suggested several principles dictating the overall productivity of compartmentalized enzymatic reactions[76]. It was suggested that the optimal enzyme density in the compartment is far below the maximal occupancy, and that each reaction compartment has a critical size, above which the productivity drops[76]. Therefore, for distributed organelles such as peroxisomes, an optimal compartmentalization strategy would be establishing multiple compartments, each under a critical size, and having less than maximal enzyme packing[76]. Our results showing that the abnormally large peroxisomes overpacking metabolic enzymes in *S. japonicus pex11Δ* cells are functionally deficient in compartmentalized amino acid biosynthesis (Fig. 3 and Fig. 4), support this hypothesis. It will be of interest to investigate if there exist the mechanism(s) controlling peroxisomal cargo import and fission to optimize relative enzyme concentrations and/or peroxisome size in various physiological situations.

In a densely packed macromolecular environment of peroxisomes, a number of considerations including reactant channelling

between biochemically linked enzymes, diffusibility of metabolic intermediates and allosteric inhibition, may all contribute to determining an optimal compartment size. Our work begins to illuminate the fundamental rules for enzymatic compartment organization, which could be applicable to other compartmentalized biochemistry in biological and man-made systems.

## Methods

### Fission yeast cell culture and transformation

Standard fission yeast growth media and conditions were used for culturing *S. japonicus* and *S. pombe*[77]. All yeast strains in this study are listed in Supplementary Data 1, and are prototrophs, unless indicated otherwise. In medium switch experiments involving growth curve measurements or image recording, cells were first grown in YES (yeast extract with L-histidine, L-leucine, adenine and uracil supplements) medium overnight to $OD_{595nm}$ 0.4–0.6 at 30 °C. Cells were briefly centrifuged and washed with either YES or EMM twice before resuspending in the desired medium at $OD_{595nm}$ 0.1–0.15. Cultures were maintained at 30 °C. Cells were sampled at indicated time points for analysis. Population growth rates were represented as the maximal doubling time calculated from optical density measurements at $OD_{595nm}$ in exponential phase cell populations. Fission yeast transformation procedure was based on an electroporation protocol described in[78].

### Genetic manipulation of fission yeasts

Molecular genetics manipulations were performed using PCR[79]- or plasmid-based homologous recombination[80]. All plasmids were built with the restriction enzyme method, using plasmids carrying *S. japonicus ura4*, *kanR* or *hphR* cloned into the pJK210-backbone, pSO550[81], pSO820[28] or pDCJ07 (gift from D. Coudreuse). mTagBFP2 was PCR-amplified from pAV0471 (gift from A. Vjestica) and cloned into pSO820 between BamHI and NotI restriction enzyme sites. pFA6a-LAP-eGFP-KanMX6 plasmid was built by inserting the LAP-eGFP sequence flanked by PacI and AscI from pSS108 (pFA6a-LAP-eGFP-His; gift from D. Teis). To build the peroxisomal targeting Gpd1[PTS1], nucleotide sequences of PTS1 derived from the last 14 amino acid residues of the *S. japonicus* Lys3 was fused in frame with the *gpd1* ORF at the C-terminus. To delocalize Gpd2 from peroxisomes, fusion PCR was carried out to remove nucleotide sequences encoding the first 28 amino acid residues from the N-terminus of Gpd2, giving rise the Gpd2[PTS2Δ]. To generate point mutations in the Gpd2 PTS2 (Gpd2[3KALG]), fusion PCR was carried out using mutagenic primers containing sequences that convert encoded lysine residues at locations 4, 8 and 18 to alanine, leucine residues at locations 9, 12 and 16 to glycine and isoleucine residue at position 26 to glycine. To generate plasmid harbouring point mutation in *pex5* ORF, fusion PCR was carried out to incorporate nucleotide sequences that mutate the tryptophan residue at position 224 to alanine (*pex5*-W224A). All proteins

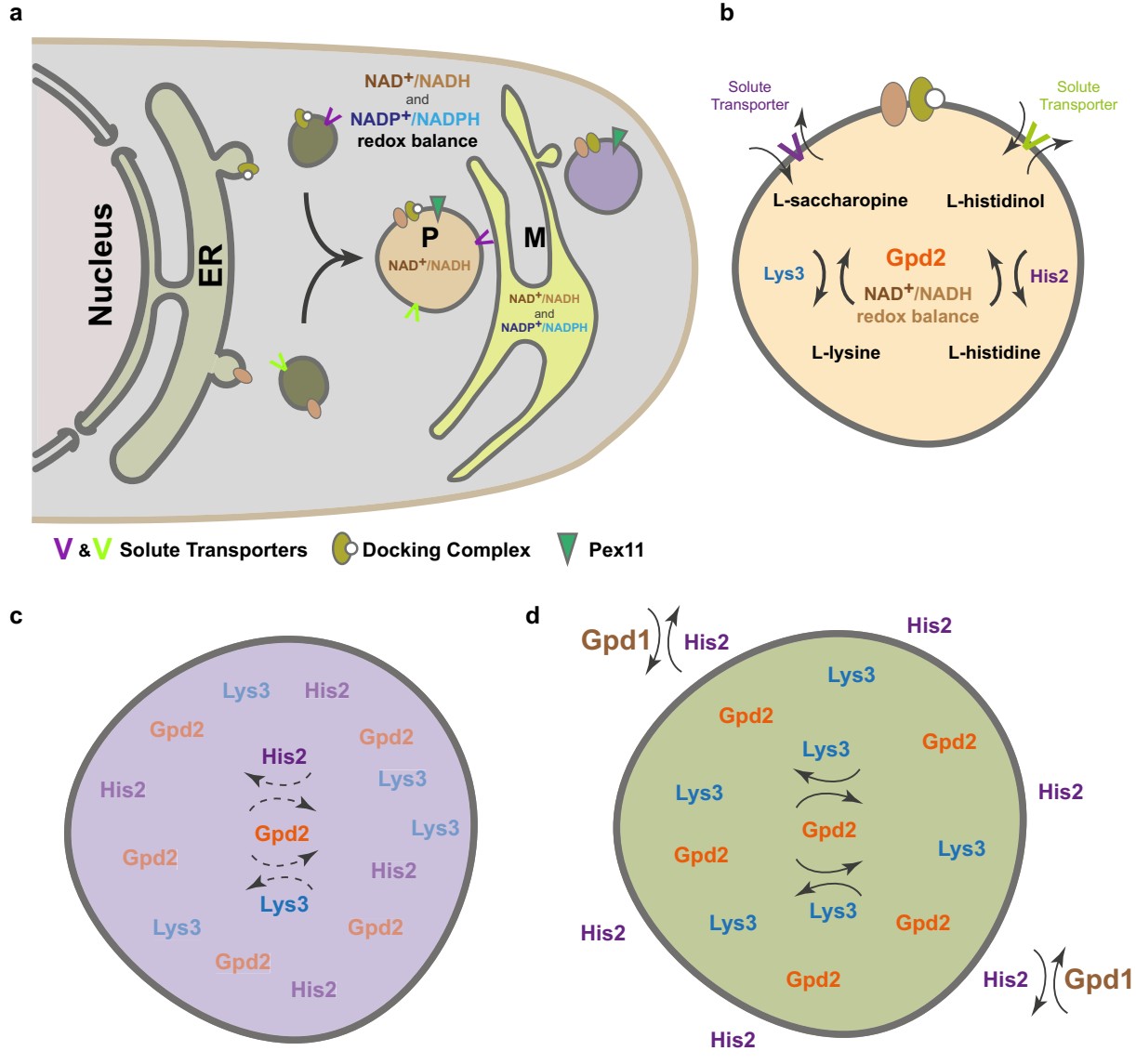

**Insufficient biosynthesis of lysine and histidine within enlarged peroxisomes**

**Relieving competition for the common NAD$^+$ cofactor restores biosynthetic capacity in enlarged peroxisomes**

**Fig. 5 | A pictorial model of intracellular redox balance facilitated by compartmentalization of amino acid biosynthesis enzymes within optimally sized peroxisomes. a** Multiple nicotinamide adenine dinucleotide NAD$^+$/NADH or nicotinamide adenine dinucleotide phosphate NADP + /NADPH-dependent metabolic reactions are present in the cytosol or compartmentalized within organelles, such as peroxisomes (P) and mitochondria (M). Peroxisome biogenesis consists of two pathways, de novo biogenesis from the endoplasmic reticulum (ER) or the fission of pre-existing peroxisomes mediated by Pex11. **b** Glycerol-3-phosphate dehydrogenase Gpd2 produces NAD$^+$ for histidine and lysine biosynthesis in peroxisomes. **c** Lysine and histidine production decrease in densely packed large peroxisomes in *pex11Δ* cells. **d** Relieving competition between compartmentalized reactions restores biosynthetic capability in enlarged peroxisomes of *pex11Δ* cells.

were expressed under their native regulatory elements, with the exception of Gpd1$^{PTS1}$, which was expressed from a *gpd2* promoter as an ectopic copy integrated into the *ura4* locus. All constructs were sequenced for verification. All primers for cloning and genotyping are listed in Supplementary Data 2.

**Microscopy**
Prior to imaging, 1 mL cell culture was concentrated to 50 μL by centrifugation at 1500 x g, 30 s. 2 μL cell suspension was loaded under a 22 × 22 mm glass coverslip (VWR, thickness: 1.5).

Spinning-disk confocal images were captured with Eclipse Ti-E inverted microscope fitted with Yokogawa CSU-X1 spinning disk confocal scanning unit, equipped with ILE 405 nm 100 mW, ILE 488 nm

50 mW and ILE OBIS 561 nm 50 mW laser lines, Nikon CFI Plan Apo Lambda 100× (N.A. = 1.45) oil objective and Andor iXon Ultra U3-888-BV monochrome EMCCD camera. Imaging was performed at 30 °C with acquisition of 13 z-slices, 0.58 μm per slice, controlled by either Andor iQ 3.6.5 (Oxford Instruments) or Andor Fusion software 2.3.0.36 (Oxford Instruments). Temperature control was achieved by the Okolab cage incubator set at 30 °C.

Epifluorescence brightfield images were acquired using Zeiss Axio Observer Z1 fluorescence microscope fitted with α Plan-FLUAR 100×/ 1.45 NA oil objective lens (Carl Zeiss) and the Orca-Flash4.0 C11440 camera (Hamamatsu). All images were taken at the medial focal plane, controlled by Zen 2012 software (blue edition, Carl Zeiss Microscopy GmbH).

## Cell morphology analysis

Measurements of cell roundness were based on datasets consisting of brightfield images taken using Zeiss Axio Observer Z1. Images were segmented using the yeast segmentation web tool YeastSpotter[82]. Individual cell contours were obtained by selecting regions of interests (ROI) from segmented cell masks and analysed in Fiji v1.53f51 using the shape descriptor function[83].

## Quantification of cellular average fluorescence intensity

Fluorescence intensity measurement of individual cells shown in Fig. 1h, Fig. 3d, Supplementary Fig. 1e and Supplementary Fig. 2g were based on datasets acquired by Andor spinning disk confocal microscope. First, all Z-slices of the brightfield channel were projected as an average intensity image. Individual cell contour was identified and separated from cell mask after segmentation of this average intensity brightfield image using YeastSpotter[82]. Second, we generated similar projections of Z-slices of fluorescence channel. Average fluorescence intensity profiles were obtained within the cell contours. Individual values from the same cell population per biological replicate were pooled, and population means of each dataset were calculated and plotted.

## Quantification of peroxisome parameters

To segment peroxisomal compartments from cells expressing various fluorescently tagged peroxisome-associated proteins, we first generated maximum intensity projections of Z-stacks of spinning disk confocal images in Fiji[83]. Individual cell contours were manually traced with the polygon selection tool. Selected whole-cell ROIs were used to crop the regions from the corresponding maximally projected fluorescence channels. Resulting individual fluorescence images were subjected to intensity thresholding to generate mask images for peroxisomes. If peroxisomes were clustered, they were separated in masks using the watershed function. Segmented peroxisomal ROIs were applied to the Z-plane projected fluorescence images of each cell. 2D-cell area and peroxisome parameters, including 2D area of individual peroxisomes and total counts of segmented entities were analysed in Fiji v1.53f51. Volume of individual peroxisomes was estimated based on the radius calculated from each 2D area measurement, assuming most peroxisomes are spherical. Peroxisome density was shown as total peroxisome counts, normalized to 2D-cell area per cell.

To measure average fluorescence intensities of peroxisomal enzymes, we applied 3D segmentation in Fiji. Briefly, Z-stacks of single cell images were analysed with 3D Object Counter v2.0 plug-in. Threshold value of 11,000, minimum size filter 1 were used for Gpd2-mNeonGreen or GFP-Lys3 labelled peroxisome segmentation. Threshold value of 10,000 and minimum size filter 1 were used for GFP-His2 labelled peroxisome segmentation.

## Amino acid quantification using GC-MS

For metabolomics experiments, cells were pre-cultured in YES overnight, washed and re-suspended in EMM and grown for 7 h till early-exponential phase at the time of harvest. The equivalent of 2 $OD_{595nm}$ was quenched by direct injection into 100% −80 °C LCMS-grade methanol (Sigma Aldrich). Polar metabolite extraction was adapted from a protocol described in ref.[84,85] Briefly, samples were extracted twice, for 15 min, on ice in LCMS-grade acetonitrile/methanol/water (2:2:1 v/v) with 1nmol *scyllo*-inositol internal standard per sample (Sigma Aldrich). Sample debris was then removed via centrifugation and polar metabolite extracts were dried using a SpeedVac Vacuum Concentrator. Dried extracts were phase-separated using −20 °C LCMS-grade chloroform/methanol/water (1:3:3, v/v) (Sigma Aldrich).

240 µl of upper, polar phase was dried into GC-MS glass vial inserts, followed by two 30 µl methanol washes. Derivatisation was performed as previously described[86] Briefly, samples were incubated overnight in 20 µl of 20 mg ml$^{-1}$ methoxyamine hydrochloride in

pyridine (Sigma Aldrich). The next day, 20 µl of N,O-bis(trimetylsilyl) trifluoroacetamide (BSTFA) and 1% trimethylchlorosilane (TMCS) (Sigma Aldrich) was added and samples were incubated at room temperature for at least 1 h before GC-MS analysis.

Metabolites were detected using Agilent 7890B-MS7000C GC-MS in EI mode as previously described[86]. Samples were injected in a random order alongside metabolite standards and regular hexane washes. Splitless injection was performed at 270 °C in a 30 m + 10 m x 0.25 mm DB-5MS + DG column (Agilent J&W) and helium was used as the carrier gas. The oven temperature was set as follows: 70 °C (2 min), gradient of 70 °C to 295 °C at 12.5 °C per minute, gradient of 295 °C to 350 °C at 25 °C per minute and a 3-min hold at 350 °C. Data was acquired using MassHunter version B.07.02.1938 software (Agilent Technologies).

Samples were analysed using a combination of MassHunter Workstation (Agilent Technologies) and MANIC, an updated version of the software GAVIN[87] for metabolite identification using retention times and mass spectra and integration of target fragment ion peaks. For abundance quantification, integrals, the known amount of scyllo-inositol internal standard (1 nmol) and the known abundances of a standardised metabolite mix run in parallel to samples (kindly gifted by Dr. James I. MacRae, Francis Crick Institute) were used to calculate an estimated nmol abundance of each metabolite in each sample. The formula used for calculating molar abundances as shown in Eqs. (1) and (2). Abundances were normalised to cell pellet weight.

$$MRRF = \frac{SI(nmol_{mm})}{met(nmol_{mm})} \times \frac{met(int_{mm})}{SI(int_{mm})} \tag{1}$$

$$met(nmol_s) = \frac{\left(\frac{met\ int_s}{SI\ int_s}\right)}{MRRF} \tag{2}$$

MRRF molar relative response factor, SI *scyllo*-inositol (internal standard), met metabolite to be quantified; mm standard metabolite mix, int integrals, s samples.

Raw metabolomics data are shown in Supplementary Data 3.

## NAD$^+$/NADH quantification

Ratios between the oxidized form NAD$^+$ and the reduced form NADH were measured using NAD$^+$/NADH quantification kit (MAK037, Sigma) with whole cell lysates of wild type *S. japonicus* cells grown under indicated conditions. Briefly, early exponential cultures were harvested by centrifugation and flash freezing in liquid nitrogen. Cell pellets were then resuspended in NAD$^+$/NADH extraction buffer with lysing matrix Y tubes (MP Biomedicals), containing 0.5 mm diameter zirconium oxide beads, before cells were lysed with the Fastprep-24 bead beating system (MP Biomedicals), Cell lysates were then filtered through a 10 kDa protein filter (Millipore) by centrifugation at 12,000 x g, 4 °C. Cleared extracts were processed as per the manufacturer's instructions.

## Protein extraction, Western blot analysis and immunodetection

5 OD equivalent of cell pellets from individual culture (OD$_{595nm}$ 0.4–0.6), were flash frozen in liquid nitrogen. Whole-cell proteins were prepared using the trichloroacetic acid (TCA) extraction protocol[88]. Briefly, frozen cell pellets were resuspended in 1 mL ice-cold water with 10% (v/v) TCA, incubated on ice for 1 h before centrifugation at 4 °C, 18,213 x g, for 8 min. Pellets were washed once with −20 °C acetone, dried at room temperature for 2 min using a SpeedVac Vacuum Concentrator (Eppendorf) before resuspended in 300 µL, 50 mM Tris-HCl, pH 7.5 buffer containing 1 mM EDTA and 1% SDS. Samples were mixed with 0.5 mm diameter zirconium oxide beads in lysing matrix Y tubes (MP Biomedicals) before lysed with the Fastprep-24 bead beating system (MP Biomedicals). Cleared cell lysates were mixed with 4x LSD

sample buffer (Invitrogen; NP 0007) containing 5% (v/v) β-mercaptoethanol, and heated at 70 °C, 10 min.

8 μL cell lysate in sample buffer was resolved on mini NuPAGE gel (Invitrogen; 4–12% gradient, Bis-Tris, 1.0 mm, NP0321BOX) and transferred to Immun-Blot LF PVDF membrane (Bio-Rad; 0.45 μm pore size, catalogue #1620264) with tank transfer method. Membranes were stained with Revert 700 total protein stain kit (Li-Cor; catalogue #926-11010) and imaged with Odyssey CLx (Li-Cor), controlled by Image Studio 5.2 (Li-Cor). After this staining procedure, membranes were blocked in Intercept TBS blocking buffer (Li-Cor; catalogue #927-60001) for 30 min before incubation with primary antibodies, mouse α-GFP (Roche, catalogue #11814460001, clone IDs 7.1 and 13.1, 1:5000) or mouse α-RFP(Chromotek, catalogue #6G6-20, clone ID 6G6, 1:2500) at 4 °C overnight. Membranes were washed three times with TBS buffer, 0.05% (v/v) Tween20, and incubated with IRDye 800CW goat α-mouse IgG secondary antibody (Li-Cor; catalogue #926-32210, 1:20,000) for 30 min, room temperature before detection with Odyssey CLx.

Protein abundance detected by immunofluorescence on membrane was analysed with Fiji v1.53f51. Average intensity of individual band within selected ROI or individual lane of total protein stain was recorded. Values of individual band intensity were normalised to their corresponding total protein contents. The resulting values of each protein in whole-cell lysate of wild type and *pex11Δ* cells grown in YES or EMM were normalised to the wild type value in YES.

### Statistics and reproducibility

All data were plotted with Prism 9 (GraphPad). *p*-values were calculated using indicated statistical functions in Prism 9. Imaging experiments shown in Figs. 1a, 1b, 1e, 1i, 3a, 3c and Supplementary Figs. 1b, 1c, 1d, 1f, 2a, 2b, 2c, 2f, 2h, 3a, 3b were repeated three times independently, and those shown in Figs. 4a, 4f and Supplementary Figs. 4a, 4b, 4c, 4g, 4k and 4l, were repeated twice independently. In all cases, the results were reproducible.

### Reporting summary

Further information on research design is available in the Nature Portfolio Reporting Summary linked to this article.

## Data availability

All data presented in graphs and uncropped scans of all blots generated in this study are included in the Source Data file. Raw metabolomics data is provided in Supplementary Data 3 file. All microscopy and Western blotting data have been deposited in the Figshare database under accession code https://doi.org/10.6084/m9.figshare.c.6798702.v1. Source data are provided with this paper.

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

## Acknowledgements

We are grateful to the Oliferenko lab and Greg Jedd for discussions and Eugene Makeyev for suggestions on the manuscript. Many thanks to James MacRae and James Ellis (Francis Crick Institute Metabolomics Science Technology Platform) for assistance with metabolomics experiments. We are grateful to Damien Coudreuse (IGDR Bordeaux), David Teis (Medical University of Innsbruck) and Aleksandar Vjestica (University of Lausanne) for sharing cloning vectors. Work in S.O. lab was supported by the Francis Crick Institute, which receives its core funding from Cancer Research UK (CC0102), the UK Medical Research Council (CC0102), and the Wellcome Trust (CC0102). This research was funded in whole or in part, by the Wellcome Trust Senior Investigator Award (103741/Z/14/Z), Wellcome Trust Investigator Award in Science (220790/Z/20/Z) and BBSRC (BB/T000481/1) to Snezhana Oliferenko. For the purpose of Open Access, the author has applied a CC-BY public copyright licence to any Author Accepted Manuscript version arising from this submission.

## Author contributions

Y.G. conceived, performed and interpreted all cell biology, genetics and physiology experiments; generated strains; analysed data; and co-wrote the manuscript. S.A. generated and analysed metabolomics data and edited the manuscript. S.O. conceived and interpreted experiments, co-wrote and edited the manuscript.

## Funding

## Competing interests

The authors declare no competing interests.
