## [Peer Review File · Nature Communications]

Peroxisomal compartmentalization of amino acid biosynthesis reactions imposes an upper limit on compartment sizeREVIEWER COMMENTS

Reviewer #1 (Remarks to the Author):

This is a very interesting paper describing how amino acid biosynthesis is compartmentalized into the peroxisomes. Through a set of well designed experiments the authors show how two key reactions in lysine and histidine biosynthesis are compartmentalized to the peroxisomes in a fission yeast species. I think the paper can be an important contribution to our understanding of how compartmentalization of reactions in eukaryotes has enabled advantages in terms of distributing e.g. balancing of redox co-factors to different compartments. I have no specific comments to the authors as I think the experimental work is done to a very high standard, but would suggest that they reflect on the following.

1) Compartmentalization will have impact on solute transport. How do the authors think this is occurring? Antiport? Proton symport? The type of transport could have implications on the energy costs, and it should at least be speculated upon.

2) Could there also be a kinetic effect by having compartmentalization. I.e. the concentration levels of NADH and NADPH (total levels) could be different in the different compartments. Also the ratio's NADH/NAD⁺ and NADPH/NADP⁺ could be different in different compartments, and this could have a significant kinetic effect. In fact it is well known that the NADH/NAD⁺ ratio is different in mitochondria and the cytosol, and a similar advantage could be used by directing some reactions that need better kinetics/thermodynamics driving forces to a different compartment. This idea is supported by the observed increased flux upon compartmentalization.

Reviewer #2 (Remarks to the Author):

The manuscript by Gu et al. describes a thorough and rigorous study on the importance of two glycerol-3-P dehydrogenases (Gpd1 and Gpd2) in *S. japonicus*. Two main conclusions emerge from this work. One regards the advantage of having compartmentalization of amino acid biosynthesis reactions. The other regards the effects of peroxisome architecture and protein concentration on metabolic fluxes. The first conclusion is nicely supported by all the data provided in the manuscript. The authors show that both dehydrogenases are required for growth of *S. japonicus* in minimum media and that a main function of Gpd2 (a peroxisomal protein) is to provide NAD⁺ to Lys3 and His2, two enzymes involved in amino acid biosynthesis. No peroxisomal Gpd2 is required when Lys3 and His2 are mislocalized to the cytosol and the levels of cytosolic Gpd are increased. However, under these conditions there is a general deregulation of amino acid metabolism, with *S. japonicus* cells accumulating large amounts of lysine. The authors conclude that the maintenance of a normal flux through the Lys biosynthesis pathway is achieved by compartmentalization of the last enzyme of the pathway (Lys3) in the peroxisome. The second conclusion focuses on very complex issues which are generally addressed using

quantitative/mathematical approaches. Here, the data on peroxisome size and protein concentration are surely interesting but, as presented and discussed, they are not so conclusive. I have some suggestions and questions:

Main issues:

The authors have quantitative data on the numbers, dimensions and relative protein concentrations of peroxisomes from the different strains. Yet all these data are described in the main text using very qualitative wording (e.g., "we observed fewer peroxisomes that were, on average, larger in pex11Δ *S. japonicus*" and "concentration of enzymes could be considerably higher"). Could the authors describe with numbers the average peroxisome in the wt, deltaPex11 and Pex5-W224A strains? If the average peroxisome in the deltaPex11 strain turns out to have a radius just ~10 % larger than that of the average peroxisome in the wt strain (as the data in Fig. 4C may suggest) then it is very difficult to understand how such a subtle difference would explain the phenotypes of the different strains.

Also, it would be important to be sure that the amounts of Gpd1, Gpd2, Lys3 and His2 per cell are similar in the different strains. This could be easily done by analyzing identical numbers of cells from the different strains by western blot. Otherwise, one can always argue that the deltaPex11 strain has a limiting amount of Gpd2.

Other issues/suggestions:

1- Lines 106-110. Could the authors provide a short explanation (or speculation) of why an increased NADH/NAD⁺ ratio is associated with oxidative stress, as inferred from the increased expression of Ctt1?

2- Fig. S2A – a fraction of Gpd2-mNG protein displays a punctate pattern in the deltaPEX5 strain. Considering that *S. japonicus* lacks PEX7 co-receptors (lines 472-472 in the manuscript), are these puncta peroxisomes? Please provide a short explanation.

3- Lines 345-354 - The finding that peroxisomes in the Pex5-W224A strain are small is very intriguing. Could the authors explain why they decided to produce a strain with this mutation?

4- Also, Pex5 is a shuttling receptor for newly synthesized peroxisomal matrix proteins with no known role in peroxisome division or membrane biogenesis. In principle, one might assume that the Pex5-W224A protein is not totally functional and that a peroxisomal enzyme(s) with some role on the peroxisomal membrane architecture is not transported efficiently to the organelle. However, as stated in lines 349-350, "import of either PTS1- or PTS2-type cargoes was not affected". How then do the authors explain the effect of the W224A mutation?

5- Was there any special reason to use the nanobody approach instead of simply deleting the PTS1 signals in His2 and Lys3? (if not, please ignore this question).

6- According to some authors, Pex11 is a membrane pore/channel (e.g., PMID: 26597702). The data presented in this manuscript show that Pex11 is not involved in the transport of DHAP/glycerol-3-P or NADH/NAD⁺.

Reviewer #3 (Remarks to the Author):

This study addresses open questions about how an organelle's size limits its function. Specifically, the authors investigate steps in histidine and lysine biosynthesis in the peroxisomes of the fission yeast *Schizosaccharomyces japonicus*. They focus on the enzymes Lys3 and His3, which are localized in peroxisomes. The main findings include compartment-specific requirements for the two glycerol-3-dehydrogenases, where Gpd1p activity is primarily in the cytoplasm and Gpd2p is required for peroxisomal conversion of NADH to NAD⁺ for amino acid synthesis. The main conceptual advance that the authors propose is the idea that peroxisome size must be maintained in an optimal range for proper function. They support this idea by showing that a mutant with large peroxisomes (*pex11Δ*) exhibits growth defects that can be suppressed by *pex5* mutations, which restore both peroxisome size and proper growth rate. This idea is interesting and nicely follows their earlier results, but it is somewhat limited by the possibility that *pex11* and *pex5* mutations alter more than just the size of peroxisomes. In general, the methodology is rigorous and well performed. Conditions are well reasoned, and imaging analysis accounts well for the variety of peroxisome size and shape. This work is of interest from both biological and biochemical perspectives. Compartmentalization is a way to increase reaction efficiency, and peroxisomes are one example of biology exploiting this for metabolic gain. The concept of an optimal upper limit for compartment size is intriguing and this study has worked to further the understanding of the biological relevance of organelle size. I have several comments/concerns that could be addressed to improve the manuscript:

1. A concern is that it remains unclear to me if peroxisome size per se is responsible for the changes in amino acid biosynthesis, or if the relevant mutations (*pex11* and *pex5*) have other effects on metabolism leading to growth/shape defects. According to SGD database, Pex11 localizes in both the ER and peroxisomes, and is listed as part of a peroxisome import complex. Doesn't this imply that the contents of peroxisomes are altered in *pex11Δ* mutants, perhaps in addition to changes in size? And are there ER changes that might also contribute to the phenotype of this mutant? Similarly, *pex5* mutants suppress cell roundness defects of *pex11* mutants, but Pex5 is likely to be altering peroxisomal content quite broadly. The conclusions about organelle size rely on these mutants, but the possibility of pleiotropic defects limits the strength of this conclusion.

2. Two suggestions related to the previous point: (a) what are the localization and levels of Lys3 and His2 in the peroxisomes of *pex11Δ* cells? It would be nice for the authors to measure these levels by microscopy, without any GBP tags in the strains. (b) Are there PTS1 or PTS2 sequences in the Lys3 or His2 sequences, in which case the authors could mutate these proteins to prevent their import into peroxisomes? Such targeted experiments would nicely complement other approaches in the paper that tend to broadly alter peroxisome import and content.

3. The authors perform a nice mass spectrometry-based quantification of amino acid levels in *pex5Δ* mutants (Figure 2). The data look convincing, but the interpretation is challenging, as many amino acid levels change in somewhat unpredictable ways. The authors give a valiant attempt at interpreting the results in the Discussion section, but it remains a source of confusion (at least for me) to connect with other results in the paper.

4. Related to the previous point: based on changes in amino acid levels in *pex5Δ*, does addition of histidine alone suppress growth rate defects of *pex5Δ* cells? It would seem that the cells have plenty of lysine, so perhaps only histidine becomes limiting?

5. Why do peroxisome defects lead to cell rounding? Many analyses in the paper rely on the strong correlations between peroxisome function, cell roundness, and cell growth rate. However, the paper might be strengthened by additional explanation or experimentation to connect these factors.

6. The microscopy images for GBP experiments (Figure 4) are very hard to assess. I would recommend showing single-channel images to complement the merged images, even if they are only in the supplemental material. I could not assess the result as presented. An alternative (or complementary?) approach would be to measure Pearson correlation coefficients for these experiments, or some other form of quantifying the colocalization. I have the same comment for colocalization experiments in Figure 1E, where some form of quantification and single channel images would be very helpful.

We are very grateful to the reviewers for their comments and suggestions. We believe that the review process has resulted in a much-improved manuscript. Please find below the individual comments followed by our responses.

Reviewer 1

This is a very interesting paper describing how amino acid biosynthesis is compartmentalized into the peroxisomes. Through a set of well designed experiments the authors show how two key reactions in lysine and histidine biosynthesis are compartmentalized to the peroxisomes in a fission yeast species. I think the paper can be an important contribution to our understanding of how compartmentalization of reactions in eukaryotes has enabled advantages in terms of distributing e.g. balancing of redox co-factors to different compartments. I have no specific comments to the authors as I think the experimental work is done to a very high standard, but would suggest that they reflect on the following.

1) Compartmentalization will have impact on solute transport. How do the authors think this is occurring? Antiport? Proton symport? The type of transport could have implications on the energy costs, and it should at least be speculated upon.

Thank you very much for your kind assessment and for prompting us to broaden the discussion of our results. We have added a small paragraph on potential costs of metabolic compartmentalization to the Discussion chapter. Briefly, it is generally agreed that many small molecules (such as glycolytic intermediates, amino acids and their precursors) may access peroxisomal interior through passive diffusion (PMID: 32806597). It remains unclear how NAD(H) is imported into peroxisomes in mammals and yeasts (PMID: 32806597), but it is possible that it could be carried in enzyme-bound form via the peroxisome protein import machinery, with its redox recycling mediated by the peroxisomal redox enzymes (PMID: 18376850 and PMID: 32174264), e.g., Gpd2 in fission yeasts.

Although this is not applicable to *S. japonicus*, the energy costs may increase with further metabolic functionality. For instance, peroxisomal FA β -oxidation, lost in fission yeasts but present in many fungi, requires dedicated ATP-dependent transporters for FA import and an antiporter promoting ATP exchange with AMP/ADP across the peroxisomal membrane (PMID: 11390660, PMID: 11566870, PMID: 33112423 and PMID: 35127709).

2) Could there also be a kinetic effect by having compartmentalization. I.e. the concentration levels of NADH and NADPH (total levels) could be different in the different compartments. Also the ratio's NADH/NAD⁺ and NADPH/NADP⁺ could be different in different compartments, and this could have a significant kinetic effect. In fact it is well known that the NADH/NAD⁺ ratio is different in mitochondria and the cytosol, and a similar advantage could be used by directing some reactions that need better kinetics/thermodynamics driving forces to a different compartment. This idea is supported by the observed increased flux upon compartmentalization.

Again, many thanks for this idea! We have added the following paragraph on how metabolic compartmentalization could increase metabolic flux to the Discussion:

“In addition to preventing a highly active enzyme from accessing the common pool of reactants, compartmentalisation may offer other benefits. Unusual for dehydrogenases, the histidinol dehydrogenase (His2 in fission yeasts) can only bind NAD⁺ after interacting with its substrate histidinol (PMID: 3307906; PMID: 28874718). It catalyses the synthesis of histidine in two sequential NAD⁺-dependent oxidations (PMID: 3307906; PMID: 28874718). The oxidation from histidinol to histidinaldehyde in the first half-reaction is relatively fast and thermodynamically unfavourable, and the sequential oxidation of histidinaldehyde is slower but irreversible (PMID: 10353847). A dedicated NADH-NAD⁺ recycling system inside peroxisomes, maintaining a high local concentration of NAD⁺ may be beneficial to such an

enzymatic reaction with suboptimal thermodynamics or kinetics by driving enzymatic flux towards production. Indeed, we have observed a reduction in histidine levels when His2 is released into the cytosol (Fig. 2).

In addition to a tailored supply of redox factors, cellular compartments may offer different proton environments for metabolic reactions (PMID: 30065243 and PMID: 35981302). For instance, at least in vitro, production of lysine and histidine by their respective dehydrogenases is favoured in alkaline pH (PMID: 17223709 and PMID: 21672513). It would be of interest to see if the pH inside fission yeast peroxisomes indeed differs from that in the cytosol, as it was shown in other systems (PMID: 17173541), and if potential pH difference could promote or constrain the metabolic flux.

Reviewer 2

The manuscript by Gu et al. describes a thorough and rigorous study on the importance of two glycerol-3-P dehydrogenases (Gpd1 and Gpd2) in *S. japonicus*. Two main conclusions emerge from this work. One regards the advantage of having compartmentalization of amino acid biosynthesis reactions. The other regards the effects of peroxisome architecture and protein concentration on metabolic fluxes. The first conclusion is nicely supported by all the data provided in the manuscript. The authors show that both dehydrogenases are required for growth of *S. japonicus* in minimum media and that a main function of Gpd2 (a peroxisomal protein) is to provide NAD⁺ to Lys3 and His2, two enzymes involved in amino acid biosynthesis. No peroxisomal Gpd2 is required when Lys3 and His2 are mislocalized to the cytosol and the levels of cytosolic Gpd are increased. However, under these conditions there is a general deregulation of amino acid metabolism, with *S. japonicus* cells accumulating large amounts of lysine. The authors conclude that the maintenance of a normal flux through the Lys biosynthesis pathway is achieved by compartmentalization of the last enzyme of the pathway (Lys3) in the peroxisome. The second conclusion focuses on very complex issues which are generally addressed using quantitative/mathematical approaches. Here, the data on peroxisome size and protein concentration are surely interesting but, as presented and discussed, they are not so conclusive. I have some suggestions and questions:

Thank you very much for your comments and suggestions!

Main issues:
The authors have quantitative data on the numbers, dimensions and relative protein concentrations of peroxisomes from the different strains. Yet all these data are described in the main text using very qualitative wording (e.g., "we observed fewer peroxisomes that were, on average, larger in *pex11Δ S. japonicus*" and "concentration of enzymes could be considerably higher").

1) Could the authors describe with numbers the average peroxisome in the wt, deltaPex11 and Pex5-W224A strains?

2) If the average peroxisome in the deltaPex11 strain turns out to have a radius just ~10 % larger than that of the average peroxisome in the wt strain (as the data in Fig. 4C may suggest) then it is very difficult to understand how such a subtle difference would explain the phenotypes of the different strains.

We now show estimated peroxisome volumes, rather than the "raw" 2D segmentation results (new Fig. 4c). As requested, we have included information on median peroxisome sizes in different genetic backgrounds in the revised manuscript. The peroxisome size distributions in the wild type and *pex11Δ* cells are clearly different, with the median peroxisome volume in EMM-grown *pex11Δ* cells being ~2.1x larger than in the wild type. Thus, the difference between the wild type and *pex11Δ* peroxisomes is not that subtle. We have also added the information on the number of peroxisomes.

3) Also, it would be important to be sure that the amounts of Gpd1, Gpd2, Lys3 and His2 per

cell are similar in the different strains. This could be easily done by analyzing identical numbers of cells from the different strains by western blot. Otherwise, one can always argue that the deltaPex11 strain has a limiting amount of Gpd2.

To address your question, we have now quantified the levels of Gpd1, Gpd2, Lys3 and His2 proteins in wild type and *pex11Δ* strains grown in YES and EMM by Western blotting (new Supplementary Fig. 4d, e). The abundances of these enzymes are similar in the wild type and *pex11Δ* cells.

Of note, the ectopic expression of GFP-Gpd1^{PTS1} in *pex11Δ* cells, driving its localization to peroxisomes, does not rescue the growth defect of *pex11Δ* cells (Fig. 3e), suggesting that the glycerol-3-phosphate dehydrogenase dosage in peroxisomes is not limiting.

Other issues/suggestions:
1- Lines 106-110. Could the authors provide a short explanation (or speculation) of why an increased NADH/NAD⁺ ratio is associated with oxidative stress, as inferred from the increased expression of Ctt1?

The way we think about it is that a high NADH/NAD⁺ ratio induces reductive stress leading to ROS production and, ultimately, the oxidative stress. One possibility is that excessive NADH build-up would provide more electrons for the NADH:ubiquinone oxidoreductase/NADH dehydrogenase (PMID: 23086143), a functional homolog of complex I in the mammalian electron transport chain. When not bound to ubiquinone, the budding yeast NADH:ubiquinone oxidoreductase Ndi1 was shown to produce H₂O₂ (PMID: 17200125; PMID: 21220430). Consistent with the idea that the NADH:ubiquinone oxidoreductases can be a source of ROS production, the budding yeast lacking either Ndi1 or Nde1 NADH dehydrogenases are more resistant to hydrogen peroxide (PMID: 31668496, PMID: 22993213). Incidentally, ubiquinone is virtually absent in *S. japonicus* – this organism is highly sensitive to oxidative stress, similar to ubiquinone-deficient *S. pombe* (PMID: 29191091). This explanation is perhaps too detailed to include in full, but we now mention the connection between increased an NADH/NAD⁺ ratio and the oxidative stress in the text.

2- Fig. S2A – a fraction of Gpd2-mNG protein displays a punctate pattern in the deltaPEX5 strain. Considering that *S. japonicus* lacks PEX7 co-receptors (lines 472-472 in the manuscript), are these puncta peroxisomes? Please provide a short explanation.

It's been suggested in the literature that the lack of Pex5 abrogates "functional" peroxisomes, although there are still peroxisome-like structures containing some membrane proteins (PMID: 29773809 and PMID: 17608706). As can be seen from the Fig. R1 below, at least a fraction of Gpd2-mNeonGreen puncta colocalizes with the transmembrane peroxin Pex14. The exact extent of colocalization is difficult to assess, given an intense cytosolic signal and abnormal peroxisome architecture in *pex5Δ* cells. However, this suggests that in principle Pex7-bound Gpd2 can translocate across the peroxisome membrane without Pex5, but Pex5 greatly enhances the targeting efficiency. We did not incorporate this figure in the manuscript, since we feel it is a bit beyond the scope of this particular story, but we can do it if you think it would be useful.

Figure R1. Wild type and *pex5*Δ *S. japonicus* cells co-expressing Gpd2-mNeonGreen (green) and Pex14-mCherry (magenta).

3- Lines 345-354 - The finding that peroxisomes in the Pex5-W224A strain are small is very intriguing. Could the authors explain why they decided to produce a strain with this mutation?

Since there has been very little information on fission yeast peroxisome architecture and function, we have constructed and phenotyped an extended series of *pex* mutants. We noticed that the W224-containing YxxxW motif of *S. japonicus* Pex5 appeared to be structurally similar to a FxxxW²⁶¹ motif of *S. cerevisiae* Pex5, which has been proposed to function in the peroxisomal import of the fatty-acyl coenzyme A oxidase Pox1, a non-canonical Pex5 cargo protein lacking a recognisable PTS1 (PMID: 11967269) and the carnitine acetyl-CoA transferase Cat2 mutant where both its N-terminal mitochondrial targeting sequence and the C-terminal PTS1 were deleted (PMID: 11967269). Although genes encoding Pox1, Cat2 and other proteins in the fatty acid oxidation pathway, are absent in the genomes of fission yeasts, we wanted to check if we could detect any peroxisomal phenotypes associated with a mutation in this motif. It was immediately apparent that in this mutant, peroxisomes were smaller and present in higher numbers. Hence, after establishing that the import and function of Lys3, His2 and Gpd2 were not affected, we have used this mutation to manipulate peroxisome size in the context of *pex11*Δ mutant. We have now added a few sentences on this mutant to the Discussion chapter.

4- Also, Pex5 is a shuttling receptor for newly synthesized peroxisomal matrix proteins with no known role in peroxisome division or membrane biogenesis. In principle, one might assume that the Pex5-W224A protein is not totally functional and that a peroxisomal enzyme(s) with some role on the peroxisomal membrane architecture is not transported efficiently to the organelle. However, as stated in lines 349-350, "import of either PTS1- or PTS2-type cargoes was not affected". How then do the authors explain the effect of the W224A mutation?

Indeed, we did not observe deficient import for the cargoes we've tested (Supplementary Fig. 4b), or detected any problems in peroxisomal "functionality" in Pex5-W224A cells. Importantly, as mentioned in the manuscript, *pex5-W224A* cells grew at wild type rates in EMM, in contrast to *pex5*Δ cells. This suggests that W224A mutation does not cause an overt loss of function phenotype in Pex5. That said, we cannot rule out that there is indeed a protein(s) important for membrane architecture/fission, transport of which could be affected in some way by the *pex5-W224A* mutation. As mentioned above, we have now added a short discussion on this mutant to the manuscript. Thank you for prompting us to do it!

5- Was there any special reason to use the nanobody approach instead of simply deleting the PTS1 signals in His2 and Lys3? (if not, please ignore this question).

Thank you for this question! We have indeed tried it. However, we discovered that removing the last few amino acids of His2 or Lys3 interferes with their enzymatic functions. Cells lacking PTS1 context in either His2 or Lys3 enzyme grow significantly slower than wild type in EMM (see Figure R2a below). In the case of His2, we directly measured the enzymatic activities of the wild type GFP-His2 and GFP-His2^{ΔPTS1}, adapting the method described in PMID: 387771 to a 96-well plate format. The proteins were isolated from cell extracts using GFP-trap magnetic beads (Chromotek). We observed complete loss of *in vitro* dehydrogenase activity when the last six amino acid residues on His2 were deleted (Figure R2b below). Furthermore, simply removing the PTS1 itself (3 amino acids) or PTS1 within the broader context (up to 13 amino acids) did not fully abrogate colocalization of Lys3 and His2 with peroxisomal marker Pex14. This could be due to bipartite localization sequences, which would be difficult to disentangle without affecting the enzyme functionality, or these enzymes piggybacking on other proteins, etc. Due to these considerations, we decided to use the nanobody approach to manipulate cellular localisation of His2 and Lys3.

Figure R2. **a** Growth rates of *S. japonicus* strains with indicated genotypes in EMM post medium switch from YES. **b** Measurement of dehydrogenase activities of GFP-tagged wild type His2 and His2^{Δlast 6a.a.}.

6- According to some authors, Pex11 is a membrane pore/channel (e.g., PMID: 26597702). The data presented in this manuscript show that Pex11 is not involved in the transport of DHAP/glycerol-3-P or NADH/NAD⁺.

Thank you! We now mention this in the Results chapter.

Reviewer 3

This study addresses open questions about how an organelle's size limits its function. Specifically, the authors investigate steps in histidine and lysine biosynthesis in the peroxisomes of the fission yeast *Schizosaccharomyces japonicus*. They focus on the enzymes Lys3 and His3, which are localized in peroxisomes. The main findings include compartment-specific requirements for the two glycerol-3-dehydrogenases, where Gpd1p activity is primarily in the cytoplasm and Gpd2p is required for peroxisomal conversion of NADH to NAD⁺ for amino acid synthesis. The main conceptual advance that the authors propose is the idea that peroxisome size must be maintained in an optimal range for proper function. They support this idea by showing that a mutant with large peroxisomes (*pex11Δ*) exhibits growth defects that can be suppressed by *pex5* mutations, which restore both peroxisome size and proper growth rate. This idea is interesting and nicely follows their earlier results, but it is somewhat limited by the possibility that *pex11* and *pex5* mutations alter more than just the size of peroxisomes. In

general, the methodology is rigorous and well performed. Conditions are well reasoned, and imaging analysis accounts well for the variety of peroxisome size and shape. This work is of interest from both biological and biochemical perspectives. Compartmentalization is a way to increase reaction efficiency, and peroxisomes are one example of biology exploiting this for metabolic gain. The concept of an optimal upper limit for compartment size is intriguing and this study has worked to further the understanding of the biological relevance of organelle size. I have several comments/concerns that could be addressed to improve the manuscript:

1. A concern is that it remains unclear to me if peroxisome size per se is responsible for the changes in amino acid biosynthesis, or if the relevant mutations (*pex11* and *pex5*) have other effects on metabolism leading to growth/shape defects. According to SGD database, Pex11 localizes in both the ER and peroxisomes, and is listed as part of a peroxisome import complex. Doesn't this imply that the contents of peroxisomes are altered in *pex11* Δ mutants, perhaps in addition to changes in size? And are there ER changes that might also contribute to the phenotype of this mutant? Similarly, *pex5* mutants suppress cell roundness defects of *pex11* mutants, but Pex5 is likely to be altering peroxisomal content quite broadly. The conclusions about organelle size rely on these mutants, but the possibility of pleiotropic defects limits the strength of this conclusion.

Thank you for your kind words and these questions and suggestions! Of course, we cannot completely rule out possible pleiotropic defects in *pex11* Δ cells. That said, we show that two types of experimental perturbations performed in *pex11* Δ background are capable of rescuing the amino acid biosynthesis phenotype. First, the introduction of the *pex5*-W224A mutation rescues both the peroxisome size and the metabolic phenotype of *pex11* Δ cells. The *pex5*-W224A mutation alone does not affect lysine or histidine biosynthesis either. Second, and perhaps more convincingly, removing either Lys3 or His2 to the cytoplasm allows the remaining reaction to occur in enlarged peroxisomes, in the absence of Pex11. This suggests that *pex11* Δ and, for that matter, *pex5*-W224A by themselves, do not interfere with the amino acid biosynthesis reactions.

In our hands *S. japonicus* Pex11-mCherry (or Pex11-GFP) expressed from its native genomic locus localizes to peroxisomes (Fig. R3a). We did not detect overlap between Pex11 and the ER marked by the luminal marker GFP-ADEL (Fig. R3b, upper panel). In the absence of Pex3, when Pex11 (and other peroxisomal membrane proteins) cannot insert in the peroxisomal membrane, Pex11 remained cytosolic (Fig. R3b, lower panel).

To address your question on ER architecture in cells lacking Pex11, we have imaged the GFP-ADEL-marked ER at both cellular mid-plane and the periphery, as we have previously done for other mutants (e.g., in PMID: 20434336). We did not observe any alterations in the overall ER organization in *pex11* Δ genetic background, as compared to the wild type (Fig. R3c).

We did not include these data in the revised manuscript, but it can be done if you think it necessary.

Figure R3. **a** Wild type *S. japonicus* cells co-expressing GFP-His2 (green) and Pex11-mCherry (magenta). Images are maximal projection of 13 z-slices. **b** Wild type and *pex3Δ* *S. japonicus* cells co-expressing the ER marker GFP-ADEL (green) and Pex11-mCherry (magenta). Images are maximal projection of two medial z-slices. **c** Wild type and *pex11Δ* *S. japonicus* cells expressing the ER marker, shown as the medial and peripheral planes.

2. Two suggestions related to the previous point: (a) what are the localization and levels of Lys3 and His2 in the peroxisomes of *pex11Δ* cells? It would be nice for the authors to measure these levels by microscopy, without any GBP tags in the strains.

As requested, we have now quantified GFP-Lys3 and GFP-His2 in the peroxisomes of wild type and *pex11Δ* cells grown in YES and EMM (new Supplementary Fig. 4c). Similar to GBP-expressing cells, the levels of both enzymes are higher in the absence of Pex11.

(b) Are there PTS1 or PTS2 sequences in the Lys3 or His2 sequences, in which case the authors could mutate these proteins to prevent their import into peroxisomes? Such targeted experiments would nicely complement other approaches in the paper that tend to broadly alter peroxisome import and content.

The Reviewer 2 asked us a similar question, so we copy our reply below.

Thank you for this question! We have indeed tried it. However, we discovered that removing the last few amino acids of His2 or Lys3 interferes with their enzymatic functions. Cells lacking PTS1 context in either His2 or Lys3 enzyme grow significantly slower than wild type in EMM (see Figure R2a below). In the case of His2, we directly measured the enzymatic activities of the wild type GFP-His2 and GFP-His2^{ΔPTS1}, adapting the method described in PMID: 387771 to a 96-well plate format. The proteins were isolated from cell extracts using GFP-trap magnetic beads (Chromotek). We observed complete loss of *in vitro* dehydrogenase activity when the last six amino acid residues on His2 were deleted (Figure R2b below). Furthermore, simply removing the PTS1 itself (3 amino acids) or PTS1 within the broader context (up to 13 amino acids) did not fully abrogate colocalization of Lys3 and His2 with peroxisomal marker Pex14. This could be due to bipartite localization sequences, which would be difficult to

disentangle without affecting the enzyme functionality, or these enzymes piggybacking on other proteins, etc. Due to these considerations, we decided to use the nanobody approach to manipulate cellular localisation of His2 and Lys3.

Figure R2. **a** Growth rates of *S. japonicus* strains with indicated genotypes in EMM post medium switch from YES. **b** Measurement of dehydrogenase activities of GFP-tagged wild type His2 and His2^{Δlast 6a.a.}.

3. The authors perform a nice mass spectrometry-based quantification of amino acid levels in *pex5Δ* mutants (Figure 2). The data look convincing, but the interpretation is challenging, as many amino acid levels change in somewhat unpredictable ways. The authors give a valiant attempt at interpreting the results in the Discussion section, but it remains a source of confusion (at least for me) to connect with other results in the paper.

We have attempted to explain these results better, both in Results and Discussion. Hopefully, it worked!

4. Related to the previous point: based on changes in amino acid levels in *pex5Δ*, does addition of histidine alone suppress growth rate defects of *pex5Δ* cells? It would seem that the cells have plenty of lysine, so perhaps only histidine becomes limiting?

It's a great question. As shown in new Supplementary Fig. 2d, although histidine improved the growth status of *pex5Δ* cells in EMM, it did not fully restore the doubling time to wild type level.

5. Why do peroxisome defects lead to cell rounding? Many analyses in the paper rely on the strong correlations between peroxisome function, cell roundness, and cell growth rate. However, the paper might be strengthened by additional explanation or experimentation to connect these factors.

We have previously shown that *S. japonicus* responds to nutrient deficiency (or premature entry into the cell cycle) by changing its cell size at division, and by scaling its geometry to the new, smaller cell size (PMID: 30664646). The geometry scaling is a morphogenetic event when 1) *S. japonicus* divides at a smaller size (shorter length), followed by 2) outgrowth of a hyperpolarized protrusion, and 3) "asymmetric" division, producing one almost spherical cell and one thinner cell with normal length-to width aspect ratio. This process is required for viability of the population, since *S. japonicus* relies on its aspect ratio to correctly position the division site. If cells cannot grow, in this case because they lack amino acids, they divide at shorter cell length and arrest as virtually spherical cells. We find this phenotype very robust

and useful in estimating population growth with a single-cell resolution. We have explained this paper in the manuscript: “The loss of cellular polarity leading to virtually spherical cells is typical for growth arrest in *S. japonicus* that relies on its geometry for cell division²⁸ and can be used as a proxy for estimating population growth with a single-cell resolution”.

6. The microscopy images for GBP experiments (Figure 4) are very hard to assess. I would recommend showing single-channel images to complement the merged images, even if they are only in the supplemental material. I could not assess the result as presented. An alternative (or complementary?) approach would be to measure Pearson correlation coefficients for these experiments, or some other form of quantifying the colocalization. I have the same comment for colocalization experiments in Figure 1E, where some form of quantification and single channel images would be very helpful.

We now show single-channel images for GBP experiments (Fig. 4f) in new Supplementary Fig. 5k. We have also added single-channel images for experiments in Fig. 1e, as new Supplementary Fig. 1c. We hope that the reviewer will agree that qualitatively, these markers colocalize.

In quantitative terms, we have estimated both Pearson’s coefficients and Mander’s coefficients for experiments in Fig. 4f and Fig. 1e, using the JaCop plug-in in Fiji. These show strong positive correlation (please see Tables R1 and R2 below). That said, there are technical limitations to this kind of analysis. Since our 3D image datasets are acquired using live cell samples, peroxisomes may move throughout the acquisition, resulting in partial displacement of the same structure between different channels. Since this issue is impossible to control, we prefer leaving the coefficient quantitations out of the paper.

Table R1. Estimated Pearson’s coefficients and Mander’s coefficients for co-localization among proteins encoded in the indicated strain backgrounds.

Strain Genotype	Pearson's Coefficient (between GFP and mCherry)	Manders' Coefficients		Pearson's Coefficient t (between GFP and mTagBFP2)	Manders' Coefficients	
		M1 (fraction of GFP overlapping mCherry)	M2 (fraction of mCherry overlapping GFP)		M1 (fraction of GFP overlapping mTagBFP2)	M2 (fraction of mTagBFP2 overlapping GFP)
GFP-lys3 tea1-mCherryGBP gpd2-mTagBFP2	0.586	0.465	0.781	0.684	0.568	0.796
pex11Δ GFP-lys3 tea1-mCherryGBP gpd2-mTagBFP2	0.607	0.612	0.817	0.732	0.724	0.882
GFP-his2 tea1-mCherryGBP gpd2-mTagBFP2	0.529	0.503	0.519	0.681	0.635	0.633
pex11Δ GFP-his2 tea1-mCherry-GBP gpd2-mTagBFP2	0.545	0.601	0.627	0.673	0.677	0.808

Table R2. Estimated Pearson’s coefficients and Mander’s coefficients for co-localization between proteins encoded in the indicated strain backgrounds.

Strain Genotype	Pearson's Coefficient (between GFP/mNG and mCherry)	Manders' Coefficients	
		M1 (fraction of GFP/mNG overlapping mCherry)	M2 (fraction of mCherry overlapping GFP/mNG)

gpd2- mNeonGreen pex14-mCherry	0.494	0.77	0.464
GFP-lys3 pex14-mCherry	0.633	0.56	0.819
GFP-his2 pex14-mCherry	0.602	0.682	0.668

REVIEWERS' COMMENTS

Reviewer #1 (Remarks to the Author):

I am very happy with the authors response to my comments and I do not have any additional comments.
Great paper

Reviewer #2 (Remarks to the Author):

The authors have addressed all my previous points in a satisfactory way. This work will be a valuable addition to the organelle/peroxisome field.

Reviewer #3 (Remarks to the Author):

The authors have done a nice job responding to reviewer comments. I do not have any additional concerns. Congratulations to the authors on a very interesting study and paper.

Reviewer 1

I am very happy with the authors response to my comments and I do not have any additional comments. Great paper.

Thank you for your kind words!

Reviewer 2

The authors have addressed all my previous points in a satisfactory way. This work will be a valuable addition to the organelle/peroxisome field.

Thank you very much for your kind comments!

Reviewer 3

The authors have done a nice job responding to reviewer comments. I do not have any additional concerns. Congratulations to the authors on a very interesting study and paper.

Thank you!